# Nonparametric serial interval estimation with uniform mixtures

**Oswaldo Gressani** [1*], **Niel Hens**[1,2]

**1** Interuniversity Institute for Biostatistics and Statistical Bioinformatics (I-BioStat), Data Science Institute, Hasselt University, Hasselt, Belgium, **2** Centre for Health Economics Research and Modelling Infectious Diseases, Vaxinfectio, University of Antwerp, Antwerp, Belgium

\* oswaldo.gressani@uhasselt.be

**Data availability statement:** Simulation results and real data applications underlying this article can be reproduced with the material provided

## Abstract

The serial interval of an infectious disease is a key instrument to understand transmission dynamics. Estimation of the serial interval distribution from illness onset data extracted from transmission pairs is challenging due to the presence of censoring and state-of-the-art methods mostly rely on parametric models. We present a fully data-driven methodology to estimate the serial interval distribution based on interval-censored serial interval data. The proposed nonparametric estimator of the cumulative distribution function of the serial interval is based on the class of uniform mixtures. Closed-form solutions are available for point estimates of different serial interval features and the bootstrap is used to construct confidence intervals. Algorithms underlying our approach are simple, stable, and computationally inexpensive, making them easily implementable in a programming language that is most familiar to a potential user. The nonparametric user-friendly routine is included in the EpiDelays package for ease of implementation. Our method complements existing parametric approaches for serial interval estimation and permits to analyze past, current, or future illness onset data streams following a set of best practices in epidemiological delay modeling.

## Author summary

Epidemiological delay distributions play a key role in outbreak analyses and in modeling infectious diseases. The serial interval is the time from illness onset in a primary case to illness onset in a secondary case and ranks among the most important delay quantities as it can be used to infer transmission patterns in mathematical and statistical models. From a statistical perspective, estimation of the serial interval distribution is complicated by the fact that the exact timing of illness onset is usually unknown and the latter event is only known to have occurred between two time points; a phenomenon called interval censoring. We propose a new inferential method to estimate the serial interval distribution from interval-censored illness onset data without relying on a parametric model. The nonparametric methodology comes with a low degree of mathematical complexity

on the GitHub repository (https://github.com/oswaldogressani/Serial_interval) based on the EpiDelays package version 0.0.1 (https://github.com/oswaldogressani/EpiDelays).

**Funding:** OG and NH were supported by the VERDI project (101045989) and the ESCAPE project (101095619), funded by the European Union. Views and opinions expressed are however those of the authors only and do not necessarily reflect those of the European Union or European Health and Digital Executive Agency (HADEA). Neither the European Union nor the granting authority can be held responsible for them. OG and NH acknowledge the financial support of the Fondation Universitaire de Belgique (file nr. AS-0608). OG and NH were also supported by the BE-PIN project (contract nr. TD/231/BE-PIN) funded by BELSPO (Belgian Science Policy Office) as part of the POST-COVID programme. The funders had no role in study design, data collection and analysis, decision to publish, or preparation of the manuscript.

**Competing interests:** The authors have declared that no competing interests exist.

and the underlying algorithms are simple, fast and stable. A user-friendly routine written in the R programming language is available in the EpiDelays package. The proposed data-driven method accounts for a set of best practices in epidemiological delay modeling and can be used to obtain point estimates and confidence intervals for often reported serial interval features.

## 1. Introduction

The serial interval (SI) is an epidemiological delay characterizing a duration between two well-defined events related to a disease. It represents the time between symptom onset in a primary case or infector and symptom onset in a secondary case or infectee [1]. This time delay can be negative as nothing restrains the infectee to experience symptom onset earlier than the infector [2]. In the literature, this interval is also known as the clinical onset serial interval [3,4]. Epidemiological and biological factors are responsible for introducing variation in times between primary and secondary events [5], so that serial intervals can be represented by a time delay distribution [6]. A different, but closely related delay quantity is the generation interval, which is defined as the duration between infection events in an infector-infectee pair [7]. The timing of an infection event is typically less likely to be observed than the timing of a symptom event and it is common practice to approximate the distribution of generation times by the SI distribution [8,9]. Serving as a proxy for generation intervals, serial intervals can be used as an instrument to measure the time scale of disease transmission [10] and are therefore key in linking the epidemic growth rate with the reproduction number [11,12]. The crucial role played by the serial interval distribution in disease transmission models emphasizes the need to have reliable, stable, and replicable statistical methodologies to estimate this quantity. Ideally, these methodologies should also follow best practices recently described in [5].

Different methods exist to estimate the distribution and features of the serial interval of an infectious disease based on data. When time intervals of illness onset between infectors and infectees are observed, the data is considered as a random sample from the population. In that case, essential features of the serial interval are estimated by either directly computing summary statistics from empirical serial intervals (e.g. mean, median, standard deviation) or by fitting a parametric distribution to observed data [13,14]. Parametric methods are by far the most common and usually include the Lognormal, Weibull, Gamma or Gaussian distributions [15–20]. For instance, a systematic review and meta-analysis of serial interval estimates for COVID-19 [21] shows that a majority of studies rely on parametric models with a frequent use of Gamma and Gaussian distributions. Estimation of model parameters is typically carried out with the maximum likelihood principle or by using the Bayesian approach, and is often based on a relatively small sample size. To our knowledge, only few attempts have been made in applying nonparametric methods to serial interval data analysis. For instance, [3] compute a nonparametric estimate of the cumulative distribution function of the serial interval of influenza based on the method of [22] to see whether different parametric models are in agreement with it, and [23] use the nonparametric bootstrap to compute confidence intervals for the clinical onset SI of SARS-CoV-2.

By definition, serial intervals involve transmission pairs. It means that a minimal requirement for SI estimation is to have data on symptom onset times for the infector and infectee. Such data can be extracted from contact tracing programmes, which permit to gain knowledge about who infected whom and provide information on timings of symptoms in infector-infectee pairs [24,25]. Commonly, serial interval data are interval censored in that only lower

and upper limits of illness onset timing is observed. This characteristic adds a layer of complexity to the estimation problem. If censoring concerns either the infector or infectee, data are said to be single interval-censored; and if censoring affects both actors in the transmission pair, data are called doubly interval-censored [26]. Thinking from a continuous time perspective, serial interval data is more often than not doubly interval-censored due to the time resolution of reporting. When the time resolution for reporting illness onset is a calendar day (as is often the case), then censoring is inherent to the calendar day, i.e. the precise timing of illness onset within the reported calendar day remains unknown. Therefore, even if exact calendar dates are observed, it is good practice to still consider the data as doubly interval-censored [5].

Despite the large number of studies conducted on the serial interval of different pathogens, most methods are difficult or impossible to reproduce in the sense that independent researchers are confronted with serious difficulties in reusing existing procedures to new data [27]. The field of infectious disease modeling suffers from alarmingly low computational reproducibility rates [28], which hinders applicability and misaligns with pandemic preparedness objectives. This reproducibility conundrum has several causes. For instance, recent meta-epidemiological surveys found that very few publications share code or data [29,30]. Other potential causes are code incompleteness and complex dependencies among multiple scripts without clear guidelines regarding computation order [28]. The study of [31] highlights that finding evidence supporting frequently cited serial interval values in the literature is a challenging task.

Hopefully, more applicable tools and methods have recently emerged to estimate epidemiological delay distributions. Originally developed for estimation of incubation period distributions, the methodology of [26] is available in an R software package [32] and associated routines are embedded in the EpiEstim package of [33] to estimate the serial interval [34]. [31] reanalyze published serial interval data on different respiratory infections by using a common statistical method and provide R code and data sets for reproducibility. The epidist package [35] and the primarycensored package [36] are also operational for serial interval estimation and account for censoring and truncation. These tools rely on parametric methods imposing distributional assumptions on the serial interval distribution and leave no room for data-driven inference.

In an attempt to complement the above-mentioned parametric methods, we develop a nonparametric approach to estimate the serial interval distribution based on illness onset data. The proposed method is entirely data-driven and applicable on a wide range of serial interval data commonly analyzed in the literature. Its chief merits are its mathematical and computational simplicity. The proposed method also aligns with some of the best practices recommended by [5], namely: (1) adjusting for double interval censoring, (2) reporting guidelines for epidemiological delays, (3) accounting for negative serial intervals and (4) reproducibility guidelines. Since R is among the most popular programming languages used in the infectious disease modeling community [28,37], the code underlying our non-parametric methodology is written in the R language and available in the EpiDelays package (https://github.com/oswaldogressani/EpiDelays). Source code comes in a lightweight format and spans only a few lines. It can thus be easily translated in another programming language if needed (e.g. Python or C++).

Next, we present our nonparametric estimator and briefly discuss some of its theoretical properties. An entire section is dedicated to simulations in order to assess the performance of our data-driven approach. Applications to transmission pair data extracted from previous outbreaks for a diverse set of pathogens underlines the wide, general, and straightforward

applicability of our method. The article concludes with a discussion surrounding different aspects of the proposed nonparametric methodology for serial interval estimation.

## 2. Methods

### 2.1. The coarse structure of serial interval data

Datasets used to estimate serial interval features are usually obtained from line list information collected during epidemics [10,34,38]. The structure is such that a line in the data list conveys information about calendar dates for the infector and infectee [31,39]. Calendar dates are not well-suited for statistical analysis. Therefore, conversion from calendar time to analysis time is carried out through a mapping from the set of calendar dates to a set of real numbers, and more commonly to a set of integers. The precise calendar date of symptom appearance may be unknown and this uncertainty translates into a range of reported dates. In that case, serial interval data are referred to as coarse data following the terminology of [40] in the sense that the timing of symptom onset is only observed to lie within a time interval; a feature also known as interval censoring. Even when precise dates are reported, there is still uncertainty with respect to the exact timing of symptom onset within a day. As such, a calendar day can be coarsened to an interval of two consecutive calendar dates, where the reported day is the lower bound and the following day is the upper bound of the interval. This means that serial interval data are usually treated as doubly interval-censored [26], i.e. the data contain a range of symptom onset dates for each primary and secondary case. After conversion of calendar time to analysis time, denote by $\overrightarrow{t_i}$ the illness onset time of the infector and by $t_i$ the illness onset time of the infectee in the $i$th transmission pair. These quantities are treated as interval-censored because the precise symptom onset time within a day is usually unknown. Let $\overrightarrow{t_{iL}}$ and $\overrightarrow{t_{iR}}$ denote the observed left and right bound, respectively, of the symptom onset time of the infector in the $i$th transmission pair and assume $\overrightarrow{t_{iL}} < \overrightarrow{t_{iR}} < +\infty$. For the infectee, a similar notation is used and we assume $t_{iL} < t_{iR} < +\infty$. The four time points $\left(\overrightarrow{t_{iL}}, \overrightarrow{t_{iR}}, t_{iL}, t_{iR}\right)$ can be used to compute the earliest possible SI time $s_{iL} = t_{iL} - \overrightarrow{t_{iR}}$ and the latest possible SI time $s_{iR} = t_{iR} - \overrightarrow{t_{iL}}$. The SI window width $s_{iR} - s_{iL} = \left(\overrightarrow{t_{iR}} - \overrightarrow{t_{iL}}\right) + (t_{iR} - t_{iL}) > 0$ sheds light upon the degree of coarseness associated with the (unobserved) serial interval in the $i$th infector-infectee pair. A schematic representation of SI data and its underlying coarseness is shown in Fig 1.

To address uncertainty about who infected whom in outbreak data, robust likelihood-based methods can be employed that explicitly account for missing or potentially incorrect transmission links (see e.g. [41]). These methods estimate the probability of transmission between individuals using contact information and the timing of symptom onset. An iterative approach is then used to reconstruct plausible transmission trees and to estimate key epidemiological delay distributions, while simultaneously identifying and mitigating the influence of unreliable data points. An alternative and simplified approach to mitigate the risk of misspecifying the infector is to concentrate on a subset of transmission pairs for which there is reasonable evidence about who the infector is [42], although this could introduce bias in the analysis.

### 2.2. A uniform mixture model

Let $\mathcal{S}$ be a real-valued random variable representing the serial interval of an infectious disease and denote by $F_{\mathcal{S}}(\cdot)$ the cumulative distribution function (cdf) of $\mathcal{S}$ with $F_{\mathcal{S}}(s) = P(\mathcal{S} \leq s) \; \forall s \in \mathbb{R}$. We adopt Laplace's principle of insufficient reason [43,44] and assume that the interval-censored SI variable of the $i$th transmission pair is uniformly distributed over the

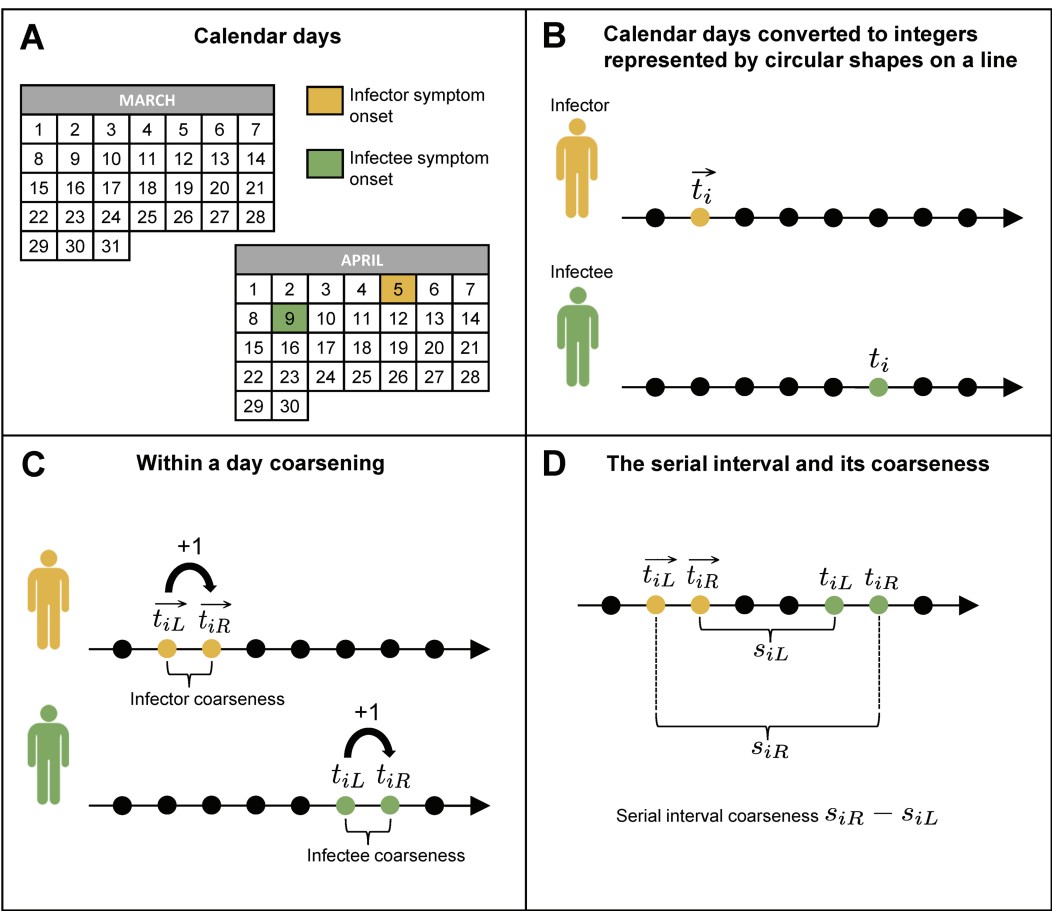

**Fig 1. Schematic representation of the coarse structure of serial interval data.** (A) The timings of symptom onset in infector-infectee pairs are usually reported as calendar dates. (B) Conversion from calendar time to analysis time is done through a mapping from reported calendar dates to a set of numbers (usually integers). (C) To account for the uncertainty in the timing of symptom onset within a day when a single calendar day is reported by the infector or infectee, a one-day coarsening of the data is implemented by constructing an interval with endpoints corresponding to two numbers resulting from the mapping of two successive calendar days in analysis time. (D) Coarseness at the serial interval level is obtained by taking the difference between the right serial interval bound $s_{iR}$ and the left serial interval bound $s_{iL}$.

censoring interval with endpoints $s_{iL}$ and $s_{iR}$, i.e. $\mathcal{S}_i \sim U(s_{iL}, s_{iR})$. The resulting cdf associated with $\mathcal{S}_i$ is denoted by:

$$\widehat{F}_{\mathcal{S}_i}(s) = \left( \frac{s - s_{iL}}{s_{iR} - s_{iL}} \right) \mathbb{I}(s_{iL} \leq s \leq s_{iR}) + \mathbb{I}(s > s_{iR}),$$

where $\mathbb{I}(\cdot)$ is the indicator function. The ordered pair $\mathcal{D}_i = (s_{iL}, s_{iR})$ denotes the $i$th transmission pair SI window constructed from the observed data points $s_{iL}$ and $s_{iR}$. Also, let $\mathcal{I} = \{\mathcal{D}_1, \dots, \mathcal{D}_n\}$ denote the set of ordered pairs representing the information set (or set of observables) constructed from serial interval data with $n$ transmission pairs. Following previous work on mixtures of uniform distributions (see e.g. [45,46]), we propose to estimate $F_{\mathcal{S}}(\cdot)$ by the $n$-component mixture $\widehat{F}_{\mathcal{S}}(s) = \sum_{i=1}^{n} \omega_i \widehat{F}_{\mathcal{S}_i}(s)$ with weights $\omega_i = n^{-1}$ for $i = 1, \dots, n$. The resulting data-driven estimate is:

$$\widehat{F}_{\mathcal{S}}(s) = \frac{1}{n} \sum_{i=1}^{n} \left\{ \left( \frac{s - s_{iL}}{s_{iR} - s_{iL}} \right) \mathbb{I}(s_{iL} \leq s \leq s_{iR}) + \mathbb{I}(s > s_{iR}) \right\}. \tag{1}$$

The above estimate is a finite convex combination of continuous functions $\widehat{F}_{\mathcal{S}_i}$ and is therefore itself a continuous function in $\mathbb{R}$. Moreover, it is a non-decreasing function since it essentially accumulates probability mass over intervals when moving along the real line in the positive direction. It is also easy to verify that $\lim_{s \to -\infty} \widehat{F}_{\mathcal{S}}(s) = 0$ and $\lim_{s \to +\infty} \widehat{F}_{\mathcal{S}}(s) = 1$, so that $\widehat{F}_{\mathcal{S}}$ is a *bona fide* cdf. Note also that $\widehat{F}_{\mathcal{S}}$ is a piecewise-linear function with breakpoints or "bends" arising at observed data points. Piecewise-linear cumulative distribution functions are endowed with interesting properties that have for instance been studied in [47,48]. These properties will guide us in computing point estimates of different serial interval features.

In parametric approaches, it is customary to work with the estimated probability density function (pdf) of the serial interval distribution, while our methodology concentrates around the estimated cumulative distribution function $\widehat{F}_{\mathcal{S}}$. This implies no loss of generality as the cdf gives a complete description of the underlying target distribution. For instance, our method can be used to compute an estimate of the basic reproduction number $\mathcal{R}_0$, the average number of secondary cases generated by a primary case in a fully susceptible population [49]. The generation interval distribution provides a link between the exponential growth rate of an epidemic $r$ and the basic reproduction number via the Lotka-Euler equation [11], namely $\mathcal{R}_0 = \left( \int_0^{+\infty} \exp(-ra) f_{\mathcal{G}}(a) \mathrm{d}a \right)^{-1}$, where $f_{\mathcal{G}}$ is the pdf of the generation interval. Using the serial interval as a proxy for the generation interval, the latter equation becomes $\mathcal{R}_0 = \left( \int_{-\infty}^{+\infty} \exp(-ra) f_{\mathcal{S}}(a) \mathrm{d}a \right)^{-1}$, where $f_{\mathcal{S}}$ is the pdf of the serial interval. Relying on the Riemann-Stieltjes integral notation, the estimated basic reproduction number using our nonparametric method is $\widehat{\mathcal{R}}_0 = \left( \int_{-\infty}^{+\infty} \exp(-ra) \mathrm{d}\widehat{F}_{\mathcal{S}}(a) \right)^{-1}$, where the integral can be solved numerically. An alternative way to proceed in estimating $\mathcal{R}_0$ without entirely leveraging our nonparametric cdf estimate is to work with a classic parametric distribution for the generation time $f_{\mathcal{G}}$ and use a parameterization that aligns with our nonparametric estimate of the mean and variance of the serial interval.

## 2.3. Point estimation

The uniform mixture model in (1) is mathematically appealing as it permits to compute frequently reported point estimates of features of the SI distribution in closed form based on the information set $\mathcal{I}$. Using the Riemann-Stieltjes integral representation of the expected value, point estimates of the SI mean $\mathbb{E}(\mathcal{S}) = \mu_{\mathcal{S}}$ and standard deviation $\sqrt{\mathbb{V}(\mathcal{S})} = \sigma_{\mathcal{S}}$ are given by:

$$
\begin{aligned}
\widehat{\mu}_{\mathcal{S}} \;=\; & \widehat{\mathbb{E}}(\mathcal{S}) = \frac{1}{n} \sum_{i=1}^{n} \left( \int_{\mathbb{R}} s \, \mathrm{d}\widehat{F}_{\mathcal{S}_i}(s) \right) = \frac{1}{n} \sum_{i=1}^{n} \widehat{\mathbb{E}}(\mathcal{S}_i) = \frac{1}{n} \sum_{i=1}^{n} \frac{(s_{iL} + s_{iR})}{2}, \\[2mm]
\widehat{\sigma}_{\mathcal{S}} \;=\; & \left[ \widehat{\mathbb{E}}(\mathcal{S}^2) - \widehat{\mu}_{\mathcal{S}}^2 \right]^{1/2} = \left[ \frac{1}{n} \sum_{i=1}^{n} \left( \int_{\mathbb{R}} s^2 \mathrm{d}\widehat{F}_{\mathcal{S}_i}(s) \right) - \widehat{\mu}_{\mathcal{S}}^2 \right]^{1/2} \\[2mm]
\;=\; & \left[ \frac{1}{n} \sum_{i=1}^{n} \widehat{\mathbb{E}}(\mathcal{S}_i^2) - \widehat{\mu}_{\mathcal{S}}^2 \right]^{1/2} \\[2mm]
\;=\; & \left[ \frac{1}{n} \sum_{i=1}^{n} \frac{(s_{iL}^2 + s_{iL} s_{iR} + s_{iR}^2)}{3} - \left( \frac{1}{n} \sum_{i=1}^{n} \frac{(s_{iL} + s_{iR})}{2} \right)^2 \right]^{1/2}.
\end{aligned}
$$

The estimated quantile function of the random variable $\mathcal{S}$ is $\widehat{q}_p(\mathcal{S}) = \inf\{s \in \mathbb{R} : \widehat{F}_{\mathcal{S}}(s) \geq p\}$, where $\widehat{q}_p(\mathcal{S})$ is the $p$-quantile of $\mathcal{S}$ for a given $p \in (0, 1)$. Denote by $\widetilde{s}_1$ and $\widetilde{s}_2$ two neighboring breakpoints of $\widehat{F}_{\mathcal{S}}$ satisfying $\widetilde{s}_1 < \widetilde{s}_2$. When $p$ is such that $p = \widehat{F}_{\mathcal{S}}(\widetilde{s}_1) = \widehat{F}_{\mathcal{S}}(\widetilde{s}_2)$, then $\widehat{F}_{\mathcal{S}}$ has a flat behavior between $\widetilde{s}_1$ and $\widetilde{s}_2$ and so $\widehat{q}_p(\mathcal{S}) = \widetilde{s}_1$ by definition of the quantile function. When $p$ is such that $\widehat{F}_{\mathcal{S}}(\widetilde{s}_1) < p < \widehat{F}_{\mathcal{S}}(\widetilde{s}_2)$, the piecewise-linear property of $\widehat{F}_{\mathcal{S}}$ can be used to compute the desired estimated quantile $\widehat{q}_p(\mathcal{S})$. In fact, by piecewise linearity, the slope of $\widehat{F}_{\mathcal{S}}$ between $\widetilde{s}_1$ and $\widehat{q}_p(\mathcal{S})$ is equal to the slope of $\widehat{F}_{\mathcal{S}}$ between $\widehat{q}_p(\mathcal{S})$ and $\widetilde{s}_2$. This allows to write an equation that can be solved for the single unknown $\widehat{q}_p(\mathcal{S})$. Mathematically:

$$\frac{\widehat{F}_{\mathcal{S}}(\widehat{q}_p(\mathcal{S})) - \widehat{F}_{\mathcal{S}}(\widetilde{s}_1)}{\widehat{q}_p(\mathcal{S}) - \widetilde{s}_1} = \frac{\widehat{F}_{\mathcal{S}}(\widetilde{s}_2) - \widehat{F}_{\mathcal{S}}(\widehat{q}_p(\mathcal{S}))}{\widetilde{s}_2 - \widehat{q}_p(\mathcal{S})}. \tag{2}$$

Solving (2) for $\widehat{q}_p(\mathcal{S})$ yields:

$$\widehat{q}_p(\mathcal{S}) = \left(\widehat{F}_{\mathcal{S}}(\widetilde{s}_2) - \widehat{F}_{\mathcal{S}}(\widetilde{s}_1)\right)^{-1} \left(\widetilde{s}_1\left(\widehat{F}_{\mathcal{S}}(\widetilde{s}_2) - p\right) + \widetilde{s}_2\left(p - \widehat{F}_{\mathcal{S}}(\widetilde{s}_1)\right)\right). \tag{3}$$

If $p$ satisfies $\widehat{F}_{\mathcal{S}}(\widetilde{s}_1) = p < \widehat{F}_{\mathcal{S}}(\widetilde{s}_2)$, then $\widehat{q}_p(\mathcal{S}) = \widetilde{s}_1$ and if $\widehat{F}_{\mathcal{S}}(\widetilde{s}_1) < p = \widehat{F}_{\mathcal{S}}(\widetilde{s}_2)$, then $\widehat{q}_p(\mathcal{S}) = \widetilde{s}_2$.

## 2.4. Quantification of uncertainty

The generic notation $\theta$ is used to represent a given feature of the SI distribution, for instance $\theta = \mu_{\mathcal{S}}$ if the mean is of interest or $\theta = \sigma_{\mathcal{S}}$ if the focus is on the standard deviation of $\mathcal{S}$. The bootstrap method will be used to compute measures of accuracy associated with the estimate $\widehat{\theta}$ [50,51]. Let $\mathcal{I}^* = \{\mathcal{D}_1^*, \ldots, \mathcal{D}_n^*\}$ denote a bootstrap sample obtained by sampling randomly and with equal probability $n$ transmission pair SI windows with replacement from $\mathcal{I}$. With $n = 4$ transmission pairs, a possible realization is $\mathcal{D}_1^* = (s_{2L}, s_{2R}), \mathcal{D}_2^* = (s_{4L}, s_{4R}), \mathcal{D}_3^* = (s_{1L}, s_{1R}), \mathcal{D}_4^* = (s_{2L}, s_{2R})$ and the bootstrap sample $\mathcal{I}^*$ is simply a set of $n$ ordered pairs. For the features of $\mathcal{S}$ presented in Sect 2.3, the bootstrap replication of $\widehat{\theta}$ denoted by $\widehat{\theta}^*$ can easily be computed based on $\mathcal{I}^*$. Generating $B$ bootstrap samples and computing their corresponding bootstrap replicate of $\widehat{\theta}$ gives access to $\mathcal{B}_{\widehat{\theta}^*} = \left\{\widehat{\theta}^{*(1)}, \ldots, \widehat{\theta}^{*(B)}\right\}$, which characterizes the bootstrap distribution of the statistic $\widehat{\theta}$. The bootstrap estimate of the standard error of $\widehat{\theta}$ can be used as a measure of accuracy of the estimate $\widehat{\theta}$. It corresponds to the empirical standard deviation of the values in $\mathcal{B}_{\widehat{\theta}^*}$:

$$\widehat{\text{se}}(\widehat{\theta}) = \left[\frac{1}{(B-1)}\sum_{b=1}^{B}\left(\widehat{\theta}^{*(b)} - \left(\frac{1}{B}\sum_{b=1}^{B}\widehat{\theta}^{*(b)}\right)\right)^2\right]^{1/2}.$$

A confidence interval for $\theta$ can be constructed from the empirical quantiles of the sample of bootstrap estimates in $\mathcal{B}_{\widehat{\theta}^*}$. Let $\widehat{\theta}_{\alpha/2}^*$ and $\widehat{\theta}_{1-\alpha/2}^*$ denote the $\alpha/2$ and $1-\alpha/2$ sample quantiles of the values in $\mathcal{B}_{\widehat{\theta}^*}$. Most software has readily available routines to compute these quantiles (e.g. the *quantile* function in R). The $100(1-\alpha)\%$ confidence interval for $\theta$ using the quantile method is denoted by $\text{CI}_{1-\alpha}(\theta) = \left[\widehat{\theta}_{\alpha/2}^*, \widehat{\theta}_{1-\alpha/2}^*\right]$. Following [52], we recommend using a bootstrap sample size of at least $B = 2000$ for confidence interval construction.

## 3. Results

### 3.1. Simulations

**3.1.1. Generating mechanism for artificial serial interval data.** To simulate artificial serial interval data, we assume that the target SI has a $\mathcal{N}(\mu_S, \sigma_S^2)$ distribution with mean $\mu_S$ and standard deviation $\sigma_S$. At the transmission pair level, the interval-censoring mechanism is governed by a discrete random variable $C$ with values $c_l = l$ for $l = 1, \ldots, L$ and probability mass function $\mathbb{P}(C = c_l) = p_l$ with $\sum_{l=1}^{L} p_l = 1$. Given a set of parameters $\mu_S, \sigma_S, L$ and $\{p_l\}_{l=1}^{L}$, a complete dataset for $n$ transmission pairs is obtained by repeating the following four steps $n$ times. 1. Draw $S$ from a $\mathcal{N}(\mu_S, \sigma_S^2)$ distribution. 2. Sample $C$ based on the chosen distribution. 3. Draw $\mathcal{U}$ from a uniform distribution $U(0,1)$. 4. Compute the left bound $S_L$ and right bound $S_R$ of the SI window of a transmission pair as $S_L = \lfloor (S - \mathcal{U}C) \rfloor$ and $S_R = \lceil (S + (1 - \mathcal{U})C) \rceil$, where $\lfloor \cdot \rfloor$ is the floor function returning the greatest integer less than or equal to its argument and $\lceil \cdot \rceil$ is the ceiling function returning the smallest integer greater than or equal to its argument. The distribution of $C$ controls the degree of data coarseness, i.e. the width of the generated serial interval windows. This simple mechanism permits to simulate frequently encountered serial interval data in the epidemiologic literature and properly takes into account the uncertainty regarding the timing of symptoms onset within the day. Said differently, for the infector and infectee, symptoms onset are only known to lie between two successive calendar days so that transmission pair data are doubly interval-censored. Mathematically this means that, under the common mapping of calendar dates to integers, infector coarseness $\overrightarrow{t_{iR}} - \overrightarrow{t_{iL}}$ and infectee coarseness $t_{iR} - t_{iL}$ are both bounded below by one. This implies that SI coarseness measured by $s_{iR} - s_{iL} = (t_{iR} - \overrightarrow{t_{iL}}) - (t_{iL} - \overrightarrow{t_{iR}}) = (\overrightarrow{t_{iR}} - \overrightarrow{t_{iL}}) + (t_{iR} - t_{iL})$ is bounded below by two. Fixing $c_1 = 1$ in our data generating mechanism ensures that the SI window $s_{iR} - s_{iL}$ is at least equal to two days. Fig 2 illustrates two sets of simulated serial interval data with $n = 15$, $\mu_S = 2.5$, $\sigma_S = 3$ and censoring distribution $p_1 = 0.80$, $p_2 = 0.15$, $p_3 = 0.05$.

**3.1.2. First set of simulations.** The performance of our nonparametric method is first assessed by assuming two target SI distributions, namely a SI distribution inspired from the SARS-CoV-2 Omicron variant $S \sim \mathcal{N}(\mu_S = 2.8, \sigma_S^2 = 2.5^2)$ [20] and a SI distribution that imitates results obtained for smallpox $S \sim \mathcal{N}(\mu_S = 16.7, \sigma_S^2 = 3.3^2)$ [31]. The distribution for the

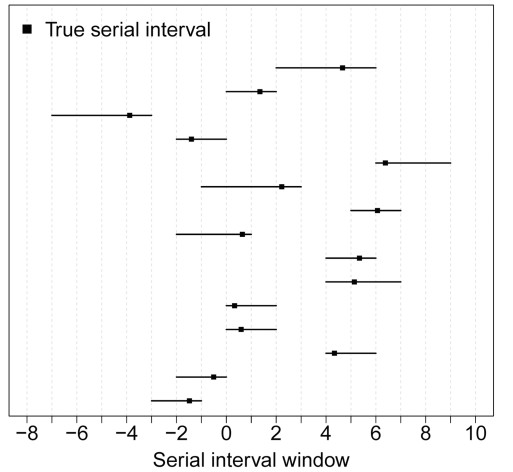
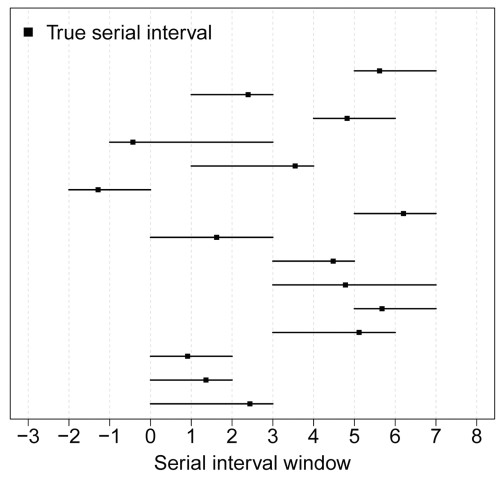

**Fig 2. Example of two coarse SI datasets of size $n = 15$ obtained with our data generating mechanism using $\mu_S = 2.5$, $\sigma_S = 3$ and the censoring distribution $p_1 = 0.80$, $p_2 = 0.15$, $p_3 = 0.05$.**

censoring mechanism is given by $p_1 = 0.80$, $p_2 = 0.15$, and $p_3 = 0.05$, so that generated SI window widths vary between 2 and 4 days. For each target SI distribution, we simulate $M = 1000$ datasets with four different sample sizes $n \in \{10, 20, 50, 100\}$; covering frequently encountered numbers of transmission pairs in the literature [21,53]; yielding a total of $2 \times 4 = 8$ scenarios. The performance of our nonparametric approach is assessed on the following often reported features of the SI distribution: mean $\mu_S$, standard deviation $\sigma_S$ and quantiles $q_{0.05}$, $q_{0.25}$ $q_{0.50}$, $q_{0.75}$, $q_{0.95}$. We use bias, empirical standard error (ESE), root mean squared error (RMSE), coverage probability of 90% and 95% confidence intervals and median confidence interval width as performance criteria (formulas of these criteria are provided in S1 Text). Confidence intervals are constructed based on $B = 2000$ bootstrap samples.

Results for Scenarios 1-4 with underlying SARS-CoV-2 Omicron-like target SI distribution are shown in Table 1. Overall, our nonparametric method based on uniform mixtures exhibits fairly good performance with relatively low bias. The coverage of confidence intervals for all the chosen SI features are satisfactorily close to their nominal level starting from $n = 50$. Under smaller sample sizes, confidence intervals for the selected SI features tend to undercover, yet results for the mean and median remain reasonable given the underlying SI coarseness of at least two days and the small number of transmission pairs. The ESE, RMSE and width of confidence intervals tend to decrease as the sample size increases. It is also worth mentioning that estimation of remote quantiles, i.e. $q_{0.05}$ and $q_{0.95}$ is more challenging and the bias for these features is usually higher. Results for Scenarios 5-8 with a smallpox-like target SI distribution are given in Table 2 and the interpretation is the same as for Scenarios 1-4 with an overall good performance of our data-driven approach for all the considered SI features. Further simulations with higher average coarseness (Scenarios S1-S3) and scenarios with a Gamma target SI distribution inspired from measles (Scenarios S4-S7) are provided in S1 Text.

**3.1.3. The impact of coarseness.**  To illustrate the negative impact of coarseness on estimates of certain SI features, we consider a target serial interval distribution $S \sim \mathcal{N}(\mu_S = 2.1, \sigma_S^2 = 1.2^2)$ inspired from influenza A [31] and run four simulation scenarios (Scenarios 9-12) with censoring distribution $p_1 = 0.80$, $p_2 = 0.15$, $p_3 = 0.05$ and sample size $n \in \{10, 20, 50, 100\}$. Results shown in Table 3 reveal how estimation performance is impacted by working with data having a coarseness degree of at least two days (i.e. doubly interval-censored SI data). Except for the mean and median, estimates of the chosen SI features tend to suffer from a larger bias as compared to the previous scenarios (Scenarios 1-8). This is because the standard deviation of the assumed influenza A serial interval target distribution $\sigma_S = 1.2$ is smaller than its counterpart in the SARS-CoV-2 Omicron ($\sigma_S = 2.5$) and smallpox ($\sigma_S = 3.3$) settings. Such a small standard deviation coupled with a degree of coarseness of at least two days for the serial interval windows blurs the information conveyed by the variation of the true (and unobserved) serial interval realizations. Said differently, the degree of coarseness "dominates" or hides the rather small variations of the true (unobserved) SI values around the mean $\mu_S$. The price to pay for such a degree of coarseness is a larger bias and lower coverage, especially for estimates of $\sigma_S$ and $q_{0.05}$, $q_{0.25}$, $q_{0.75}$, $q_{0.95}$ as shown in Table 3.

To further stress the role played by the degree of coarseness in SI data, we consider the same influenza A setting but with a hypothetical coarseness degree that is close to zero. In particular, after generating $S$, we assume that the left bound of the serial interval window is $S_L = S - \varepsilon/2$ and that the right bound is $S_R = S + \varepsilon/2$ with $\varepsilon = 0.01$ so that the degree of coarseness of the SI window is equal to $\varepsilon$ and we refer to this censoring scheme as $\varepsilon$-coarseness. Simulations under $\varepsilon$-coarseness for the influenza A setting are implemented for a sample size $n \in \{10, 20, 50, 100\}$, yielding Scenarios 13-16 and results are shown in Table 4. Without surprise,

**Table 1. Results for Scenarios 1-4 with $M = 1000$ simulated datasets, censoring distribution $(p_1 = 0.8, p_2 = 0.15, p_3 = 0.05)$, $n \in \{10, 20, 50, 100\}$ and target $\mathcal{S} \sim \mathcal{N}(\mu_{\mathcal{S}} = 2.8, \sigma_{\mathcal{S}}^2 = 2.5^2)$ inspired from [20] that imitates the SI distribution of the SARS-CoV-2 Omicron variant. The first column contains the selected features of $\mathcal{S}$, namely the mean, standard deviation, 5th, 25th, 50th, 75th and 95th quantiles. Bias, ESE, RMSE, coverage probability (CP) and median confidence interval width ($\Delta$CI) are used as performance criteria.**

| Scenario 1 ($n = 10$) | Bias | ESE | RMSE | $CP_{90\%}$ | $CP_{95\%}$ | $\Delta CI_{90\%}$ | $\Delta CI_{95\%}$ |
|---|---|---|---|---|---|---|---|
| $\mu_{\mathcal{S}}$ | 0.043 | 0.794 | 0.794 | 86.60 | 91.90 | 2.400 | 2.850 |
| $\sigma_{\mathcal{S}}$ | -0.039 | 0.533 | 0.534 | 74.90 | 79.50 | 1.393 | 1.637 |
| $q_{0.05}$ | 0.066 | 1.493 | 1.494 | 64.20 | 65.10 | 2.500 | 2.833 |
| $q_{0.25}$ | 0.000 | 0.954 | 0.954 | 83.90 | 88.80 | 2.961 | 3.341 |
| $q_{0.50}$ | 0.017 | 0.892 | 0.892 | 86.90 | 93.40 | 2.750 | 3.401 |
| $q_{0.75}$ | 0.081 | 0.944 | 0.947 | 87.20 | 91.30 | 3.120 | 3.502 |
| $q_{0.95}$ | 0.073 | 1.420 | 1.421 | 57.30 | 59.10 | 2.667 | 2.906 |
| Scenario 2 ($n = 20$) | Bias | ESE | RMSE | $CP_{90\%}$ | $CP_{95\%}$ | $\Delta CI_{90\%}$ | $\Delta CI_{95\%}$ |
| $\mu_{\mathcal{S}}$ | 0.016 | 0.562 | 0.562 | 88.00 | 93.10 | 1.800 | 2.126 |
| $\sigma_{\mathcal{S}}$ | 0.050 | 0.390 | 0.393 | 83.50 | 87.90 | 1.097 | 1.298 |
| $q_{0.05}$ | -0.210 | 1.060 | 1.080 | 82.70 | 84.60 | 2.333 | 2.619 |
| $q_{0.25}$ | -0.056 | 0.677 | 0.679 | 88.80 | 93.10 | 2.199 | 2.600 |
| $q_{0.50}$ | 0.001 | 0.646 | 0.645 | 88.60 | 94.40 | 2.033 | 2.442 |
| $q_{0.75}$ | 0.098 | 0.715 | 0.721 | 87.10 | 92.40 | 2.155 | 2.615 |
| $q_{0.95}$ | 0.103 | 0.981 | 0.986 | 78.60 | 81.00 | 2.333 | 2.667 |
| Scenario 3 ($n = 50$) | Bias | ESE | RMSE | $CP_{90\%}$ | $CP_{95\%}$ | $\Delta CI_{90\%}$ | $\Delta CI_{95\%}$ |
| $\mu_{\mathcal{S}}$ | -0.005 | 0.348 | 0.348 | 89.10 | 93.50 | 1.150 | 1.375 |
| $\sigma_{\mathcal{S}}$ | 0.086 | 0.251 | 0.265 | 87.30 | 93.50 | 0.748 | 0.890 |
| $q_{0.05}$ | -0.174 | 0.645 | 0.668 | 88.40 | 92.80 | 1.963 | 2.240 |
| $q_{0.25}$ | -0.075 | 0.431 | 0.437 | 89.00 | 94.50 | 1.395 | 1.684 |
| $q_{0.50}$ | 0.002 | 0.404 | 0.404 | 89.70 | 94.30 | 1.303 | 1.552 |
| $q_{0.75}$ | 0.070 | 0.434 | 0.439 | 89.00 | 94.50 | 1.389 | 1.656 |
| $q_{0.95}$ | 0.152 | 0.673 | 0.690 | 84.30 | 89.00 | 1.875 | 2.151 |
| Scenario 4 ($n = 100$) | Bias | ESE | RMSE | $CP_{90\%}$ | $CP_{95\%}$ | $\Delta CI_{90\%}$ | $\Delta CI_{95\%}$ |
| $\mu_{\mathcal{S}}$ | 0.013 | 0.254 | 0.254 | 89.60 | 94.80 | 0.835 | 0.990 |
| $\sigma_{\mathcal{S}}$ | 0.114 | 0.168 | 0.203 | 86.50 | 93.50 | 0.550 | 0.653 |
| $q_{0.05}$ | -0.175 | 0.453 | 0.486 | 87.10 | 93.20 | 1.408 | 1.701 |
| $q_{0.25}$ | -0.076 | 0.316 | 0.325 | 89.00 | 93.90 | 1.005 | 1.198 |
| $q_{0.50}$ | 0.015 | 0.296 | 0.296 | 89.90 | 94.60 | 0.941 | 1.125 |
| $q_{0.75}$ | 0.099 | 0.309 | 0.325 | 87.90 | 94.00 | 1.005 | 1.195 |
| $q_{0.95}$ | 0.214 | 0.465 | 0.512 | 85.80 | 93.40 | 1.413 | 1.704 |

when coarseness is virtually zero, our nonparametric method shows good performance with negligible bias and confidence intervals that tend to have close to nominal coverage values starting from $n = 50$ for all the considered SI features.

**3.1.4. A note on asymptotic bias.** Coarseness is responsible for introducing bias in estimates of SI features. As can be seen from Scenarios 1-12, when $n$ increases, the bias does not necessarily decrease. This is because the sample size considered in these scenarios is not large enough to fully reveal how the underlying degree of coarseness impacts the estimates. To show the "asymptotic" impact of coarseness, we run simulations for the SARS-CoV-2 Omicron, smallpox and influenza A settings with $n = 500$. Results are shown in S1 Text (Scenarios S8-S10) and reveal good performance for the mean and median but undercoverage for the standard deviation and quantiles depending on the setting. To further highlight the asymptotic bias argument, we compare how estimates provided by our data-driven approach evolve with sample size when assuming a degree of coarseness of at least two days and when considering the hypothetical case of $\varepsilon$-coarseness in the SARS-CoV-2 Omicron setting. For a sequence of sample sizes between $n = 6$ and $n = 500$, we compute $M = 50$ estimates for each

**Table 2. Results for Scenarios 5-8 with $M = 1000$ simulated datasets, censoring distribution $(p_1 = 0.8, p_2 = 0.15, p_3 = 0.05)$, $n \in \{10, 20, 50, 100\}$ and target $\mathcal{S} \sim \mathcal{N}(\mu_{\mathcal{S}} = 16.7, \sigma_{\mathcal{S}}^2 = 3.3^2)$ inspired from [31] that imitates the SI distribution of smallpox. The first column contains the selected features of $\mathcal{S}$, namely the mean, standard deviation, 5th, 25th, 50th, 75th and 95th quantiles. Bias, ESE, RMSE, coverage probability (CP) and median confidence interval width ($\Delta$CI) are used as performance criteria.**

| Scenario 5 ($n = 10$) | Bias | ESE | RMSE | CP$_{90\%}$ | CP$_{95\%}$ | $\Delta$CI$_{90\%}$ | $\Delta$CI$_{95\%}$ |
|---|---|---|---|---|---|---|---|
| $\mu_{\mathcal{S}}$ | -0.025 | 1.065 | 1.065 | 85.90 | 90.50 | 3.150 | 3.701 |
| $\sigma_{\mathcal{S}}$ | -0.137 | 0.713 | 0.726 | 72.10 | 77.10 | 1.849 | 2.175 |
| $q_{0.05}$ | 0.163 | 1.913 | 1.919 | 48.20 | 51.00 | 3.167 | 3.667 |
| $q_{0.25}$ | -0.054 | 1.313 | 1.314 | 86.40 | 90.50 | 4.204 | 4.663 |
| $q_{0.50}$ | -0.043 | 1.253 | 1.253 | 86.00 | 92.90 | 3.600 | 4.592 |
| $q_{0.75}$ | -0.005 | 1.352 | 1.351 | 86.50 | 88.80 | 4.250 | 4.655 |
| $q_{0.95}$ | -0.187 | 1.866 | 1.875 | 54.50 | 55.30 | 3.167 | 3.667 |
| Scenario 6 ($n = 20$) | Bias | ESE | RMSE | CP$_{90\%}$ | CP$_{95\%}$ | $\Delta$CI$_{90\%}$ | $\Delta$CI$_{95\%}$ |
| $\mu_{\mathcal{S}}$ | 0.011 | 0.750 | 0.750 | 87.60 | 93.30 | 2.325 | 2.775 |
| $\sigma_{\mathcal{S}}$ | -0.035 | 0.495 | 0.496 | 80.50 | 85.20 | 1.447 | 1.714 |
| $q_{0.05}$ | -0.212 | 1.483 | 1.497 | 67.60 | 70.10 | 2.969 | 3.381 |
| $q_{0.25}$ | -0.019 | 0.905 | 0.904 | 90.20 | 94.30 | 2.911 | 3.500 |
| $q_{0.50}$ | 0.022 | 0.867 | 0.866 | 88.80 | 94.20 | 2.667 | 3.227 |
| $q_{0.75}$ | 0.039 | 0.951 | 0.952 | 86.90 | 92.20 | 2.863 | 3.447 |
| $q_{0.95}$ | -0.067 | 1.337 | 1.338 | 75.40 | 76.60 | 3.200 | 3.538 |
| Scenario 7 ($n = 50$) | Bias | ESE | RMSE | CP$_{90\%}$ | CP$_{95\%}$ | $\Delta$CI$_{90\%}$ | $\Delta$CI$_{95\%}$ |
| $\mu_{\mathcal{S}}$ | -0.003 | 0.476 | 0.475 | 89.00 | 94.20 | 1.520 | 1.810 |
| $\sigma_{\mathcal{S}}$ | 0.036 | 0.326 | 0.328 | 88.10 | 92.00 | 0.990 | 1.173 |
| $q_{0.05}$ | -0.085 | 0.890 | 0.893 | 85.00 | 89.50 | 2.641 | 3.000 |
| $q_{0.25}$ | -0.062 | 0.601 | 0.604 | 89.30 | 95.30 | 1.865 | 2.227 |
| $q_{0.50}$ | 0.001 | 0.531 | 0.531 | 89.10 | 94.30 | 1.742 | 2.088 |
| $q_{0.75}$ | 0.034 | 0.587 | 0.588 | 88.10 | 93.60 | 1.876 | 2.246 |
| $q_{0.95}$ | 0.096 | 0.896 | 0.901 | 86.20 | 89.00 | 2.732 | 3.061 |
| Scenario 8 ($n = 100$) | Bias | ESE | RMSE | CP$_{90\%}$ | CP$_{95\%}$ | $\Delta$CI$_{90\%}$ | $\Delta$CI$_{95\%}$ |
| $\mu_{\mathcal{S}}$ | -0.008 | 0.329 | 0.329 | 89.90 | 95.20 | 1.088 | 1.290 |
| $\sigma_{\mathcal{S}}$ | 0.076 | 0.226 | 0.238 | 89.40 | 94.40 | 0.734 | 0.869 |
| $q_{0.05}$ | -0.146 | 0.612 | 0.629 | 88.60 | 94.30 | 1.933 | 2.291 |
| $q_{0.25}$ | -0.065 | 0.412 | 0.417 | 89.70 | 94.90 | 1.345 | 1.602 |
| $q_{0.50}$ | -0.009 | 0.379 | 0.379 | 89.80 | 95.40 | 1.239 | 1.473 |
| $q_{0.75}$ | 0.045 | 0.410 | 0.412 | 89.70 | 95.80 | 1.338 | 1.603 |
| $q_{0.95}$ | 0.148 | 0.638 | 0.655 | 88.50 | 93.80 | 1.992 | 2.353 |

selected SI feature and analyze how the mean estimate evolves with $n$. Fig 3 shows results for the SARS-CoV-2 Omicron target SI distribution with a degree of coarseness of at least two days (panel A) and under the hypothetical case of $\varepsilon$-coarseness (panel B). A coarseness degree of at least two days introduces bias in our estimates. This is especially visible for the standard deviation and quantiles $q_{0.05}$ and $q_{0.95}$. This bias reaches a limit as $n$ grows large and is an unavoidable facet of coarseness that negatively impacts the confidence interval coverage performance. Under the hypothetical $\varepsilon$-coarseness setting, our estimates exhibit good performance and stabilize around the true SI features as $n$ increases.

## 3.2. Applications

To further validate our nonparametric method, we consider different applications on serial interval data from past outbreaks that are publicly available. A textual analysis is provided for each individual dataset and results are summarized in Table 5.

**3.2.1. Influenza A (2009 H1N1 influenza) at a New York City school.** We start by analyzing a dataset based on illness onset dates of $n = 16$ infector-infectee pairs obtained from

**Table 3. Results for Scenarios 9-12 with $M = 1000$ simulated datasets, censoring distribution ($p_1 = 0.8, p_2 = 0.15, p_3 = 0.05$), $n \in \{10, 20, 50, 100\}$ and target $\mathcal{S} \sim \mathcal{N}(\mu_\mathcal{S} = 2.1, \sigma_\mathcal{S}^2 = 1.2^2)$ inspired from [31] that imitates the SI distribution of influenza A.** The first column contains the selected features of $\mathcal{S}$, namely the mean, standard deviation, 5th, 25th, 50th, 75th and 95th quantiles. Bias, ESE, RMSE, coverage probability (CP) and median confidence interval width ($\Delta$CI) are used as performance criteria.

| Scenario 9 ($n = 10$) | Bias | ESE | RMSE | $CP_{90\%}$ | $CP_{95\%}$ | $\Delta CI_{90\%}$ | $\Delta CI_{95\%}$ |
|---|---|---|---|---|---|---|---|
| $\mu_\mathcal{S}$ | -0.008 | 0.413 | 0.413 | 85.50 | 90.90 | 1.200 | 1.450 |
| $\sigma_\mathcal{S}$ | 0.171 | 0.261 | 0.313 | 84.40 | 90.00 | 0.629 | 0.737 |
| $q_{0.05}$ | -0.329 | 0.681 | 0.756 | 65.80 | 72.80 | 1.350 | 1.667 |
| $q_{0.25}$ | -0.150 | 0.463 | 0.487 | 83.30 | 90.00 | 1.333 | 1.595 |
| $q_{0.50}$ | -0.006 | 0.432 | 0.432 | 86.00 | 91.40 | 1.297 | 1.546 |
| $q_{0.75}$ | 0.141 | 0.470 | 0.490 | 84.20 | 90.40 | 1.333 | 1.575 |
| $q_{0.95}$ | 0.292 | 0.693 | 0.751 | 75.80 | 80.20 | 1.250 | 1.667 |
| Scenario 10 ($n = 20$) | Bias | ESE | RMSE | $CP_{90\%}$ | $CP_{95\%}$ | $\Delta CI_{90\%}$ | $\Delta CI_{95\%}$ |
| $\mu_\mathcal{S}$ | 0.006 | 0.282 | 0.282 | 89.60 | 94.30 | 0.925 | 1.100 |
| $\sigma_\mathcal{S}$ | 0.223 | 0.184 | 0.289 | 73.30 | 83.70 | 0.514 | 0.609 |
| $q_{0.05}$ | -0.383 | 0.485 | 0.618 | 79.30 | 86.00 | 1.178 | 1.371 |
| $q_{0.25}$ | -0.160 | 0.323 | 0.360 | 85.80 | 91.90 | 1.023 | 1.222 |
| $q_{0.50}$ | 0.010 | 0.298 | 0.298 | 90.00 | 94.30 | 0.975 | 1.159 |
| $q_{0.75}$ | 0.169 | 0.324 | 0.365 | 87.70 | 92.80 | 1.006 | 1.205 |
| $q_{0.95}$ | 0.380 | 0.492 | 0.622 | 84.20 | 89.50 | 1.215 | 1.400 |
| Scenario 11 ($n = 50$) | Bias | ESE | RMSE | $CP_{90\%}$ | $CP_{95\%}$ | $\Delta CI_{90\%}$ | $\Delta CI_{95\%}$ |
| $\mu_\mathcal{S}$ | -0.003 | 0.186 | 0.186 | 89.20 | 93.80 | 0.590 | 0.710 |
| $\sigma_\mathcal{S}$ | 0.242 | 0.113 | 0.267 | 31.40 | 43.30 | 0.350 | 0.415 |
| $q_{0.05}$ | -0.401 | 0.315 | 0.509 | 65.10 | 77.20 | 0.886 | 1.028 |
| $q_{0.25}$ | -0.165 | 0.195 | 0.255 | 79.30 | 87.80 | 0.658 | 0.789 |
| $q_{0.50}$ | -0.003 | 0.203 | 0.203 | 89.10 | 93.10 | 0.637 | 0.757 |
| $q_{0.75}$ | 0.159 | 0.224 | 0.275 | 81.10 | 89.40 | 0.677 | 0.801 |
| $q_{0.95}$ | 0.412 | 0.319 | 0.521 | 70.40 | 81.40 | 0.892 | 1.027 |
| Scenario 12 ($n = 100$) | Bias | ESE | RMSE | $CP_{90\%}$ | $CP_{95\%}$ | $\Delta CI_{90\%}$ | $\Delta CI_{95\%}$ |
| $\mu_\mathcal{S}$ | 0.008 | 0.134 | 0.134 | 87.70 | 93.60 | 0.420 | 0.500 |
| $\sigma_\mathcal{S}$ | 0.250 | 0.077 | 0.262 | 3.90 | 8.10 | 0.259 | 0.308 |
| $q_{0.05}$ | -0.408 | 0.242 | 0.474 | 41.60 | 54.90 | 0.707 | 0.820 |
| $q_{0.25}$ | -0.149 | 0.140 | 0.204 | 71.70 | 79.50 | 0.462 | 0.559 |
| $q_{0.50}$ | 0.016 | 0.148 | 0.149 | 87.10 | 92.80 | 0.458 | 0.541 |
| $q_{0.75}$ | 0.166 | 0.164 | 0.233 | 72.00 | 81.40 | 0.498 | 0.587 |
| $q_{0.95}$ | 0.438 | 0.222 | 0.491 | 51.40 | 64.60 | 0.703 | 0.828 |

the supplementary appendix of [15]. After fitting a Weibull distribution to the data, the authors obtain a median serial interval of 2.7 days (CI95%: 2.0-3.5) and a 95th quantile of 5.1 days (CI95%: 3.6-6.5). Our nonparametric method estimates that the median SI is 2.8 days ($\widehat{se} = 0.6$; CI95%: 1.8-3.8) and the 95th quantile estimate is 5.2 days ($\widehat{se} = 0.3$; CI95%: 4.6-5.7). Fig 4A summarizes the observed serial interval windows. Fig 4B shows the estimated cdf $\widehat{F}_\mathcal{S}$ (black curve), point estimates (dots) and 95% CIs for selected quantiles of $\mathcal{S}$.

**3.2.2. Influenza A (2009 H1N1 influenza) in San Antonio, Texas, USA.** We analyze another influenza dataset [34,54] containing doubly interval-censored serial interval data from the 2009 influenza A outbreak in San Antonio, Texas, USA [55]. Our methodology estimates the mean serial interval at 4.0 days ($\widehat{se} = 0.4$; CI95%: 3.2-4.9). The standard deviation is estimated at 1.9 days ($\widehat{se} = 0.3$; CI95%: 1.2-2.4) and the 95th quantile is at 7.8 days ($\widehat{se} = 0.8$; CI95%: 5.5-8.5). Serial interval windows and estimates of different features of $\mathcal{S}$ are shown in Fig 5.

**3.2.3. Illness onset data for COVID-19 in Wuhan, China.** [17] share data on illness onset dates of $n = 6$ infector-infectee pairs and estimate that the serial interval has a mean

**Table 4. Results for Scenarios 13-16 with $M = 1000$ simulated datasets, $\varepsilon$-coarseness with $\varepsilon = 0.01$, $n \in \{10, 20, 50, 100\}$ and target $\mathcal{S} \sim \mathcal{N}(\mu_{\mathcal{S}} = 2.1, \sigma_{\mathcal{S}}^2 = 1.2^2)$ inspired from [31] that imitates the SI distribution of influenza A. The first column contains the selected features of $\mathcal{S}$, namely the mean, standard deviation, 5th, 25th, 50th, 75th and 95th quantiles. Bias, ESE, RMSE, coverage probability (CP) and median confidence interval width ($\Delta$CI) are used as performance criteria.**

| Scenario 13 ($n = 10$) | Bias | ESE | RMSE | $CP_{90\%}$ | $CP_{95\%}$ | $\Delta CI_{90\%}$ | $\Delta CI_{95\%}$ |
|---|---|---|---|---|---|---|---|
| $\mu_{\mathcal{S}}$ | -0.011 | 0.389 | 0.389 | 84.10 | 88.90 | 1.120 | 1.333 |
| $\sigma_{\mathcal{S}}$ | -0.111 | 0.270 | 0.292 | 63.90 | 69.30 | 0.667 | 0.793 |
| $q_{0.05}$ | 0.159 | 0.704 | 0.721 | 38.00 | 38.50 | 0.915 | 1.192 |
| $q_{0.25}$ | 0.013 | 0.496 | 0.496 | 90.70 | 92.20 | 1.807 | 1.875 |
| $q_{0.50}$ | -0.152 | 0.462 | 0.486 | 81.90 | 93.50 | 1.249 | 1.902 |
| $q_{0.75}$ | -0.035 | 0.519 | 0.520 | 89.20 | 90.40 | 1.810 | 1.874 |
| $q_{0.95}$ | -0.166 | 0.743 | 0.761 | 37.90 | 38.70 | 0.933 | 1.204 |
| Scenario 14 ($n = 20$) | Bias | ESE | RMSE | $CP_{90\%}$ | $CP_{95\%}$ | $\Delta CI_{90\%}$ | $\Delta CI_{95\%}$ |
| $\mu_{\mathcal{S}}$ | 0.001 | 0.276 | 0.276 | 86.30 | 91.70 | 0.845 | 1.007 |
| $\sigma_{\mathcal{S}}$ | -0.045 | 0.188 | 0.193 | 78.30 | 82.50 | 0.536 | 0.637 |
| $q_{0.05}$ | -0.245 | 0.609 | 0.656 | 59.80 | 65.00 | 0.793 | 1.047 |
| $q_{0.25}$ | -0.081 | 0.371 | 0.379 | 83.10 | 92.80 | 1.034 | 1.428 |
| $q_{0.50}$ | -0.071 | 0.343 | 0.350 | 88.40 | 92.30 | 1.066 | 1.226 |
| $q_{0.75}$ | -0.091 | 0.364 | 0.375 | 88.50 | 89.20 | 1.217 | 1.263 |
| $q_{0.95}$ | -0.268 | 0.486 | 0.555 | 65.20 | 65.50 | 1.302 | 1.365 |
| Scenario 15 ($n = 50$) | Bias | ESE | RMSE | $CP_{90\%}$ | $CP_{95\%}$ | $\Delta CI_{90\%}$ | $\Delta CI_{95\%}$ |
| $\mu_{\mathcal{S}}$ | -0.003 | 0.169 | 0.169 | 88.50 | 94.20 | 0.549 | 0.653 |
| $\sigma_{\mathcal{S}}$ | -0.018 | 0.116 | 0.118 | 85.20 | 90.80 | 0.368 | 0.434 |
| $q_{0.05}$ | 0.018 | 0.338 | 0.338 | 90.90 | 92.20 | 1.260 | 1.345 |
| $q_{0.25}$ | 0.001 | 0.234 | 0.234 | 87.50 | 92.80 | 0.711 | 0.860 |
| $q_{0.50}$ | -0.022 | 0.208 | 0.209 | 90.80 | 94.60 | 0.695 | 0.823 |
| $q_{0.75}$ | -0.007 | 0.228 | 0.228 | 88.70 | 94.00 | 0.702 | 0.854 |
| $q_{0.95}$ | -0.024 | 0.345 | 0.346 | 89.80 | 90.90 | 1.208 | 1.282 |
| Scenario 16 ($n = 100$) | Bias | ESE | RMSE | $CP_{90\%}$ | $CP_{95\%}$ | $\Delta CI_{90\%}$ | $\Delta CI_{95\%}$ |
| $\mu_{\mathcal{S}}$ | 0.002 | 0.122 | 0.122 | 88.60 | 94.00 | 0.392 | 0.466 |
| $\sigma_{\mathcal{S}}$ | -0.006 | 0.083 | 0.083 | 87.00 | 91.90 | 0.267 | 0.317 |
| $q_{0.05}$ | -0.047 | 0.257 | 0.262 | 88.60 | 94.20 | 0.802 | 0.984 |
| $q_{0.25}$ | -0.018 | 0.168 | 0.169 | 88.20 | 94.20 | 0.523 | 0.635 |
| $q_{0.50}$ | -0.013 | 0.152 | 0.152 | 89.30 | 95.00 | 0.484 | 0.582 |
| $q_{0.75}$ | -0.010 | 0.166 | 0.166 | 89.00 | 93.80 | 0.530 | 0.625 |
| $q_{0.95}$ | -0.050 | 0.245 | 0.250 | 87.10 | 88.50 | 0.794 | 0.870 |

of 7.5 days (CI95%: 5.3-19) based on a parametric model involving a Gamma distribution. Raw data come as calendar dates of illness onset for infector-infectee pairs. We therefore apply a one-day coarsening of the data to recover the desired doubly interval-censored structure. Our nonparametric method gives a mean serial interval estimate of 6.3 days ($\widehat{se} = 0.8$; CI95%: 4.7-7.7) and a median SI of 6.7 days ($\widehat{se} = 1.0$; CI95%: 4.0-8.0).

**3.2.4. Illness onset data for COVID-19 with $n = 28$ infector-infectee pairs.** A richer serial interval dataset on COVID-19 is provided by [18]. They obtained doubly interval-censored data on $n = 28$ infector-infectee pairs and estimated features of the serial interval based on a Bayesian parametric approach. The authors estimate the median serial interval to be 4.0 days (CrI95%: 3.1-4.9), where CrI denotes the credible interval. The mean and standard deviation of the serial interval are estimated at 4.7 days (CrI95%: 3.7-6.0) and 2.9 days (CrI95%: 1.9-4.9), respectively. Our nonparametric method estimates the median serial interval at 3.8 days ($\widehat{se} = 0.4$; CI95%: 3.2-4.8). Estimates for the mean and standard deviation are 4.6 days ($\widehat{se} = 0.5$; CI95%: 3.7-5.6) and 2.6 days ($\widehat{se} = 0.3$; CI95%: 2.0-3.0), respectively. A graphical output of the nonparametric results is shown in Fig 6.

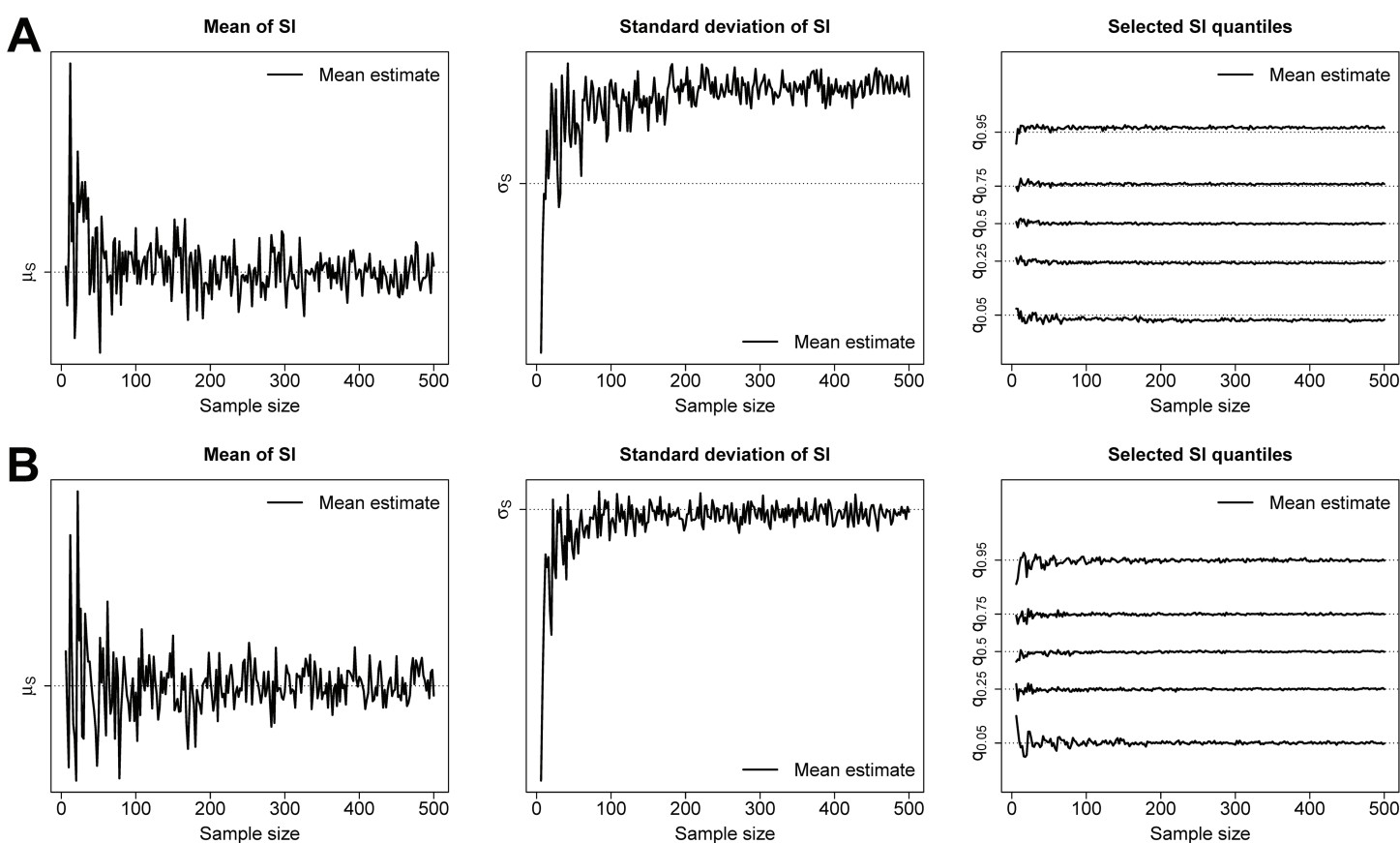

**Fig 3. Mean estimates of selected SI features computed over $M = 50$ simulated datasets for a sequence of sample sizes ranging between $n = 6$ and $n = 500$ when the underlying target SI distribution mimics the SARS-CoV-2 Omicron setting [20].** Dotted lines indicate the true value of a SI feature. (A) Serial interval coarseness of at least two days generated by the censoring distribution $p_1 = 0.8$, $p_2 = 0.15$, $p_3 = 0.05$. (B) Hypothetical $\varepsilon$-coarseness setting with $\varepsilon = 0.01$.

**3.2.5. Illness onset data for COVID-19 in Belgium.** [20] report data on illness onset dates of $n = 2161$ transmission pairs for the Omicron variant of SARS-CoV-2 and $n = 334$ infector-infectee pairs for the Delta variant. Fitting a Gaussian distribution to the data using a Bayesian approach, the authors obtain a median serial interval of 2.75 days (CrI95%: 2.65-2.86) and a standard deviation of 2.54 days (CrI95%: 2.46-2.61) for Omicron. For Delta, they obtain a median serial interval of 3.00 days (CrI95%: 2.73-3.26) and a standard deviation of 2.49 days (CrI95%: 2.31-2.69). Treating the data as doubly interval-censored, our data-driven approach estimates the median SI at 2.62 days ($\widehat{se} = 0.05$; CI95%: 2.52-2.73) and the standard deviation at 2.60 days ($\widehat{se} = 0.05$; CI95%: 2.50-2.69) for Omicron. For the Delta variant, the nonparametric approach estimates the median SI at 3.06 days ($\widehat{se} = 0.16$; CI95%: 2.73-3.35) and the estimated standard deviation is 2.54 days ($\widehat{se} = 0.09$; CI95%: 2.36-2.73).

## 4. Discussion

Our new data-driven methodology permits to estimate serial interval features based on coarse illness onset data without making parametric assumptions with respect to the SI distribution. The proposed nonparametric estimates are based on uniform mixtures and the resulting piecewise-linear structure of the cumulative distribution function allows to compute point estimates of several SI features in closed form. Such a mathematical tractability implies

**Table 5. Nonparametric estimates obtained with our method and parametric estimates of SI features (mean $\widehat{\mu}_S$, standard deviation $\widehat{\sigma}_S$, median $\widehat{q}_{0.50}$, and 95th quantile $\widehat{q}_{0.95}$) for different publicly available serial interval datasets. Values in round brackets correspond to 95% confidence intervals for our method and 95% confidence or credible intervals for parametric methods. The third column indicates the sample size. NR: Not Reported. The symbol * indicates that information was obtained by contacting the corresponding author of the article listed in the data source column.**

| Disease and data source | Method | n | $\widehat{\mu}_S$ | $\widehat{\sigma}_S$ | $\widehat{q}_{0.50}$ | $\widehat{q}_{0.95}$ |
|---|---|---|---|---|---|---|
| Influenza A [15] | Nonparametric | 16 | 2.8 (2.1-3.5) | 1.5 (1.2-2.7) | 2.8 (1.8-3.8) | 5.2 (4.6-5.7) |
| Influenza A [15] | Parametric [15] | 16 | 2.8 (NR) | 1.3 (NR) | 2.7 (2.0-3.5) | 5.1 (3.6-6.5) |
| Influenza A [54] | Nonparametric | 16 | 4.0 (3.2-4.9) | 1.9 (1.2-2.4) | 3.9 (3.1-4.7) | 7.8 (5.5-8.5) |
| COVID-19 [17] | Nonparametric | 6 | 6.3 (4.7-7.7) | 2.0 (0.9-2.6) | 6.7 (4.0-8.0) | 9.4 (7.7-9.8) |
| COVID-19 [17] | Parametric [17] | 6 | 7.5 (5.3-19.0) | NR | NR | NR |
| COVID-19 [18] | Nonparametric | 28 | 4.6 (3.7-5.6) | 2.6 (2.0-3.0) | 3.8 (3.2-4.8) | 9.6 (7.9-10.1) |
| COVID-19 [18] | Parametric [18] | 28 | 4.7 (3.7-6.0) | 2.9 (1.9-4.9) | 4.0 (3.1-4.9) | 9.8 (7.5-14.8) |
| COVID-19 Omicron [20] | Nonparametric | 2161 | 2.75 (2.65-2.86) | 2.60 (2.50-2.69) | 2.62 (2.52-2.73) | 7.32 (6.99-7.59) |
| COVID-19 Omicron [20] | Parametric [20] | 2161 | 2.75 (2.65-2.86) | 2.54 (2.46-2.61) | 2.75 (2.65-2.86) | 6.92 (6.76-7.09)* |
| COVID-19 Delta [20] | Nonparametric | 334 | 3.00 (2.72-3.26) | 2.54 (2.36-2.73) | 3.06 (2.73-3.35) | 7.13 (6.67-7.70) |
| COVID-19 Delta [20] | Parametric [20] | 334 | 3.00 (2.73-3.26) | 2.49 (2.31-2.69) | 3.00 (2.73-3.26) | 7.09 (6.70-7.53)* |

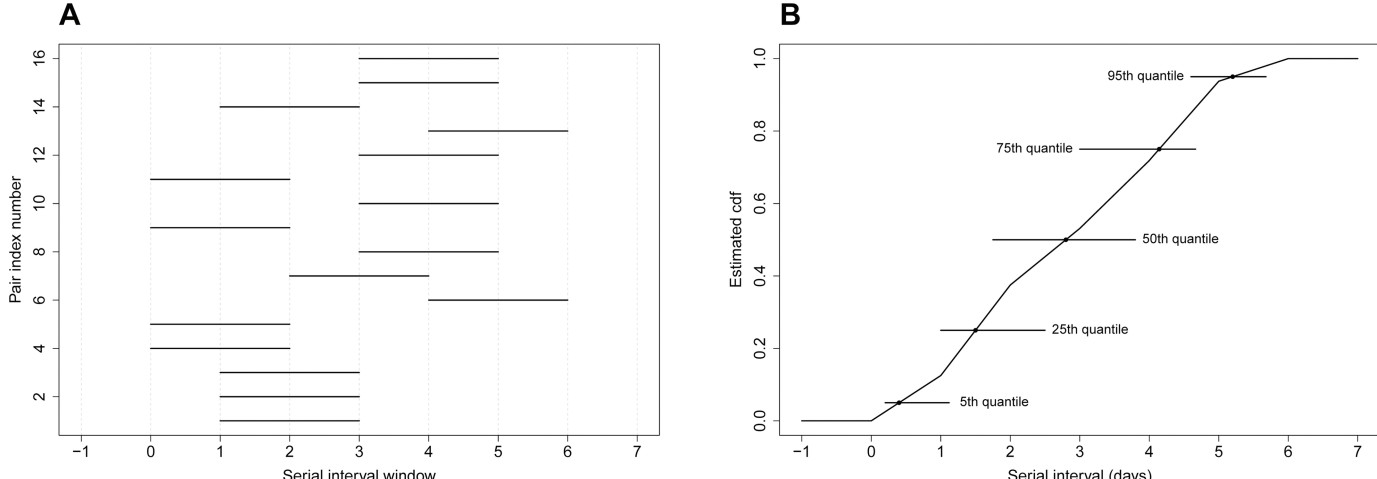

**Fig 4. (A) Serial interval windows of influenza A for *n* = 16 infector-infectee pairs at a New York City school [15].** (B) Nonparametric estimate $\widehat{F}_S$ (black curve), point estimates (dots) and 95% CIs (horizontal lines) for selected quantiles of $S$.

a low computational cost in quantifying uncertainty via the bootstrap. Simulation results suggest that the proposed nonparametric methodology will provide a reasonable approximation to the true underlying SI distribution in a large number of real-world use cases if the spread of the target distribution is not too much dominated by the degree of coarseness. A visual inspection of serial interval windows after adjusting for double interval censoring already gives an insightful assessment of whether or not coarseness dominates the spread of the underlying target SI distribution. The smaller the frequency of overlapping SI windows in one-day intervals, the richer is the signal conveying information about the spread of the underlying distribution and hence the more confident we can be in estimates of the standard

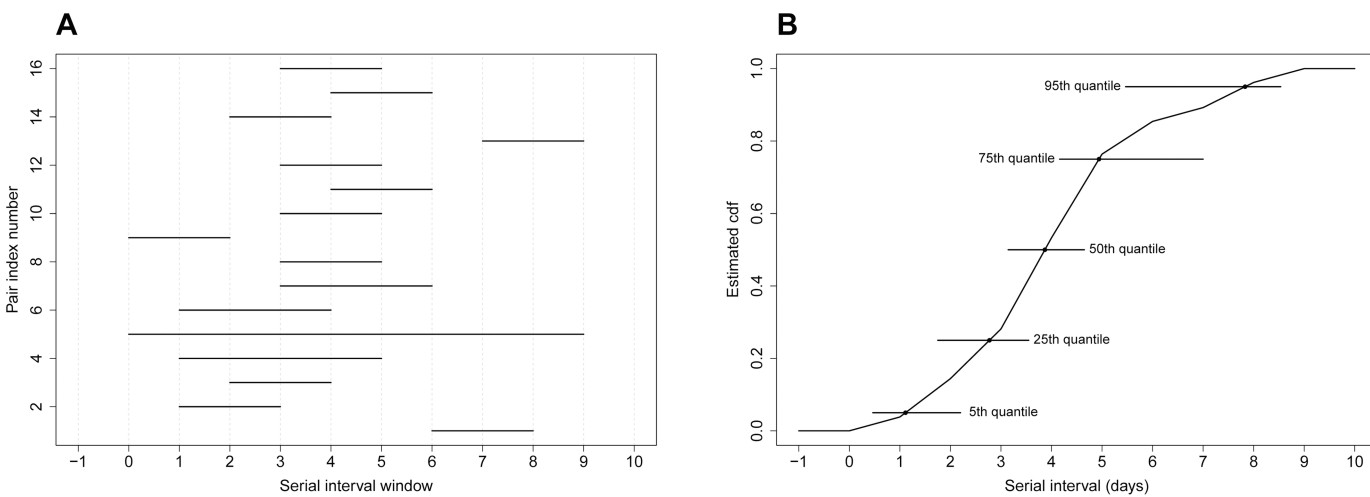

**Fig 5. (A) Serial interval windows of influenza A for *n* = 16 infector-infectee pairs in San Antonio, Texas, USA [55].** (B) Nonparametric estimate $\widehat{F}_{\mathcal{S}}$ (black curve), point estimates (dots) and 95% CIs (horizontal lines) for selected quantiles of $\mathcal{S}$.

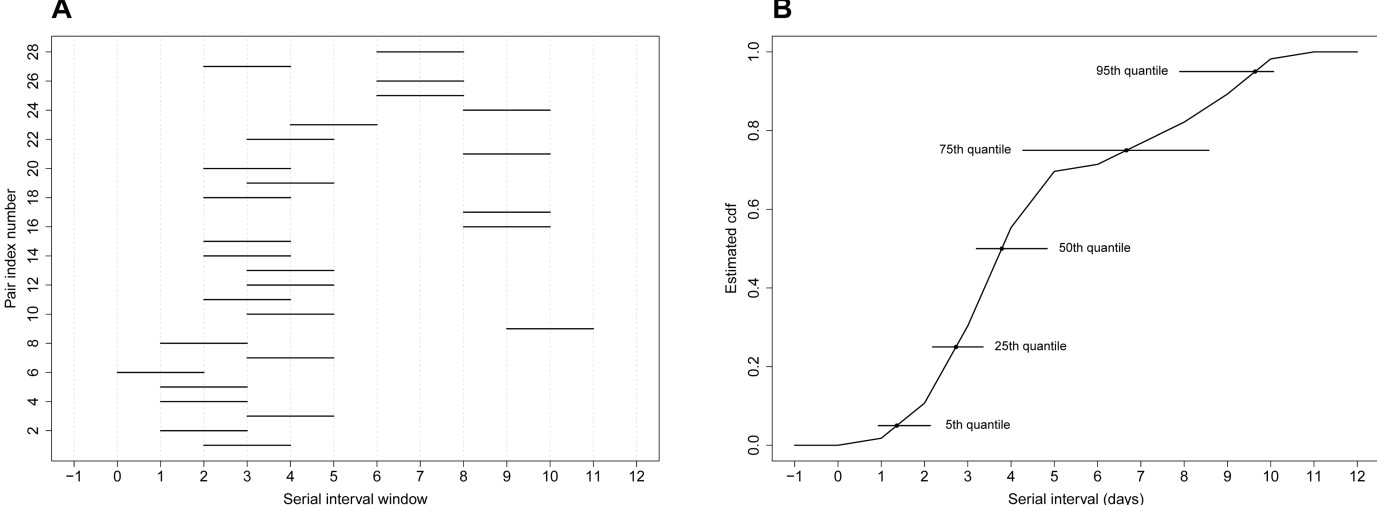

**Fig 6. (A) Serial interval windows of COVID-19 for *n* = 28 infector-infectee pairs [18].** (B) Nonparametric estimate $\widehat{F}_{\mathcal{S}}$ (black curve), point estimates (dots) and 95% CIs (horizontal lines) for selected quantiles of $\mathcal{S}$.

deviation and tail quantiles. Furthermore, we have shown that estimates of some SI features (mainly the standard deviation and tail quantiles) will remain biased even under large sample sizes due to the presence of coarseness.

While our method is specifically tailored for working with serial interval data that has been adjusted for double interval censoring, it is important to highlight that it does not adjust for right truncation. Right truncation means that SI windows are absent from the data because, at the time of data collection, the information required to build a SI window for an infector-infectee pair (i.e. two successive symptom onset times) is not yet available. The problem of right truncation appears in real-time settings and implies an overrepresentation of shorter

serial intervals, which in turn can lead to underestimation of SI statistics [5,18]. Right truncation is accentuated during the early stage of an epidemic when it undergoes a growing phase [35]. In retrospective analyses, right truncation is usually not a problem if the surveillance period is long enough to provide a representative sample [5], and in that case, our data-driven approach does not require further adjustment. Methodological developments to correct for right truncation bias in estimating serial interval distributions have only recently emerged in parametric settings [6,18,35]. An interesting future research direction would be to extend existing right truncation adjustment approaches to our nonparametric setting.

There is a surface-level similarity between the nonparametric approach proposed here and our previous work on incubation period estimation [56], however these methods are radically different in several ways. First, in our incubation period paper, we work from a Bayesian perspective and leverage the power and flexibility of Laplacian-P-splines [57,58] to estimate the incubation density. The nonparametric approach proposed here is not Bayesian and does not require the specification of a prior. Second, the distribution of incubation times is modeled in a semiparametric way and the model includes spline parameters, while our data-driven method for serial intervals is entirely parameter-free. Third, there is a non-negligible difference in terms of computational complexity. In [56], we use Markov chain Monte Carlo (MCMC) to sample from the posterior distribution of the model parameters, while here, the computational cost to obtain estimates of SI features is drastically reduced and mostly present in the resampling scheme of the bootstrap. Our nonparametric method for serial intervals is also mathematically less technical and thus perhaps more accessible to a broader set of users. For all these reasons, we believe that our approach for estimating incubation times and the newly proposed nonparametric method for estimating the serial interval distribution can be seen as complementary tools.

The proposed nonparametric method has several distinct strengths. First, being entirely data-driven, the method can be directly used to sketch the main characteristics of the SI distribution without imposing any parametric assumption. Moreover, the nonparametric estimate of the cumulative distribution function can be used as a benchmark to visually assess whether a chosen parametric model agrees with our data-driven fit, i.e. as an informal lack-of-fit test. Second, our method naturally deals with negative serial interval values and can thus be applied in a wide range of practical settings. Third, mathematical technicalities and computational complexity are kept minimal. This means that algorithms underlying our approach are very simple and can be easily translated and used in a programming language most preferred by the user. We developed a user-friendly routine for the proposed nonparametric serial interval estimation methodology that is available in the EpiDelays package (https://github.com/oswaldogressani/EpiDelays). Fourth, our method is in alignment with some of the best practices recommended by [5]. For instance, it naturally accounts for doubly interval-censored data. Also, our method automatically provides an estimate of variability (standard deviation) along with an estimate of central tendency, and these estimates are accompanied by confidence intervals via the bootstrap. Furthermore, the fact that the underlying code has a small footprint means that the method is easily reproducible. This facilitates serial interval analyses on past, current or future illness onset data streams.

A limitation of our method is that the estimated cdf obtained with uniform mixtures tells us that there is zero probability below the smallest observed left SI bound and that the serial interval lies with probability one below the largest observed right SI bound. Allowing for more flexible tails that go beyond the range of the observation set may be more realistic.

As previously mentioned, a challenging future research direction would be to adjust the proposed nonparametric approach for right truncation. Alternatively, it could be interesting to investigate how the data-driven method behaves under different weighting schemes.

Instead of attributing an equal weight of $n^{-1}$ to each serial interval window, we can for instance think of a rule that puts more weight to SI windows with smaller widths (i.e. with a lower degree of coarseness) since those windows are endowed with less uncertainty as compared to wider serial interval windows. Finally, a more theoretic study related to asymptotic properties and coarseness could provide interesting insights about the behavior of bias in our setting and give a flavor about the quality of information that can be extracted if the underlying serial interval data are characterized by an overall high degree of coarseness.

## Supporting information

**S1 Text.** Formulas of performance criteria used in the simulation study and additional simulation results.
(PDF)

## Author contributions

**Conceptualization:** Oswaldo Gressani.

**Formal analysis:** Oswaldo Gressani.

**Funding acquisition:** Niel Hens.

**Methodology:** Oswaldo Gressani.

**Software:** Oswaldo Gressani.

**Supervision:** Niel Hens.

**Validation:** Oswaldo Gressani.

**Visualization:** Oswaldo Gressani.

**Writing – original draft:** Oswaldo Gressani.

**Writing – review & editing:** Niel Hens.

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
