## [Decision Letter · Decision Letter 0]

27 Feb 2025

PCOMPBIOL-D-24-01995

Nonparametric serial interval estimation

PLOS Computational Biology

Dear Dr. Gressani,

Thank you for submitting your manuscript to PLOS Computational Biology. After careful consideration, we feel that it has merit but does not fully meet PLOS Computational Biology's publication criteria as it currently stands. Therefore, we invite you to submit a revised version of the manuscript that addresses the points raised during the review process.

Please submit your revised manuscript within 60 days Apr 29 2025 11:59PM. If you will need more time than this to complete your revisions, please reply to this message or contact the journal office at ploscompbiol@plos.org. Please include the following items when submitting your revised manuscript:

We look forward to receiving your revised manuscript.

Kind regards,

Benjamin Peirce Holder, Ph.D.

Guest Editor

PLOS Computational Biology

Denise Kühnert

Section Editor

PLOS Computational Biology

**Additional Editor Comments :**

Thank you for submitting your interesting and well-written manuscript! We apologize for the abnormal length of the review. It was difficult to assemble a panel of reviewers around the New Year, and I appreciate the work done by all reviewers.

The three reviewers were all positive in their general opinion of your manuscript. Two were supportive of a re-submission (with revisions), with one reviewer suggesting that the work did not meet the bar of originality/innovation for PLoS Computational Biology. I can understand both arguments, but I personally consider the method to be new and interesting enough (and of value to the infectious disease and modeling communities) to warrant publication, if a satisfactory resubmission is completed. Therefore, as guest editor, I support a "Major Revision".

I essentially support all of the comments and questions provided by the reviewers and would like to see them addressed in the revision. If you feel a reviewer's question or comment can be addressed in the response letter without requiring adjustment to the text, you are welcome to make the argument for that, but I think most proposed edits (or answers to questions asked) would improve the manuscript and its readability by a broad audience. A few general issues certainly warrant substantial change/additions to the text/presentation:

* Two reviewers wanted to see, within the Introduction, more context of serial interval estimation, with additional citations from the literature. It might be useful to have some of this within the Discussion/Conclusion at the end, where prior methods could be compared to those presented here.

* There are questions raised about the applicability of the serial interval instrument itself (e.g., infector/infectee discrimination, early epidemic censoring, time variability of the distribution, underlying assumptions) which can be raised in the text and acknowledged/addressed, without any change to the method and results presented.

* There are questions about the bootstrap method, which should be addressed, including the question of i.i.d. random variables and the rationale for obtaining n-element bootstrap samples. Even if answers to these are standard within the technique, it would be useful to remind the readers briefly of the basics of the bootstrap method, and how you are implementing a standard application. I personally would find it useful for you to flesh out a bit more how the samples determine the statistics quoted (mean/median and their confidence intervals). It looks to me like the originally constructed cdf does not directly provide the point estimates, but that these and the intervals are obtained from the bootstrap samples. But, in places the interval bars and point estimates don't seem to line up with the visual distribution of the plotted samples (but this may just be a display issue where density of samples can't be seen). Relatedly, it seems that the bootstrap samples are qualitatively different qualitatively than cdf from original dataset. You obtain the piecewise-linear empirical cdf from the censored data, but then you obtain the bootstrap "samples" as (I assume) step-function cdfs from n samples of the empirical pdf (by inverse-cdf method), which are then smoothed. It's not clear to me exactly how this method precisely fits with bootstrap's "resampling the data" (what is the "data" here, and what are the "samples") and I would appreciate clarification (within the text, if it is useful, or just in your response, if it is obvious!).

* The question by Reviewer 1 about how other measures (e.g., basic reproductive number) can be inferred, should be addressed. The idea that "interval estimates of any desired features of S" can be obtained in the method is clear, but some discussion of the next steps would be useful: how can it be used by modelers (or policy-makers) to calculate derived quantities or be used in applications.

* As Reviewer 2 notes, the manuscript is essentially a methodological contribution, but the "Applications" section seems to promise novel insights about the analyzed data. Perhaps this section should be something like "Model Verification". Additionally, as multiple reviewers suggested, it would be useful to have at least a summary of the simulation results from the Supplementary Material presented in the main text, as part of "verification". In the comparison with the data (or in the final Discussion/Conclusion section), please make clear the conclusions drawn: is the reader simply to see that your method essentially agrees with prior methods, or do the slightly different results indicate a "better" use of the data than parametric methods? As a methodological contribution, it would be useful (as suggested by Reviewer 2), to have at least a brief discussion of where the method might break down or give results that are at odds with other methods. If it is possible to provide "applications" (without too much additional effort) that give more than a simple verification of the technique, but extend the application to extract new insights from the data, that would certainly be useful to modelers and policy makers and could form a distinct "Applications" section.

* It would be useful to have the verification/application data summarized in a table, showing results for all data sets considered.

-BPH

**Journal Requirements:**

3) Your manuscript is missing the following sections: Results, and Discussion.  Please ensure all required sections are present and in the correct order. Make sure section heading levels are clearly indicated in the manuscript text, and limit sub-sections to 3 heading levels. An outline of the required sections can be consulted in our submission guidelines here:

5) We have noticed that you have uploaded Supporting Information files, but you have not included a list of legends. Please add a full list of legends for your Supporting Information files after the references list.

2) If the funders had no role in your study, please state: "The funders had no role in study design, data collection and analysis, decision to publish, or preparation of the manuscript.".

**Reviewers' comments:**

Reviewer's Responses to Questions

Reviewer #1: This manuscript aims to provide a method to estimate the serial interval of an infectious disease using a nonparamtric method. The serial interval is a key parameter to quantify disease transmission, and has important applications to inform disease intervention. First, it is a measure of the epidemic time scale and thus mathematically links the exponential growth rate to the basic reproduction number. This method is commonly used in the early stage of an epidemic, when the exponential growth rate is one of the disease parameters that can be reliably estimated. Secondly, it is commonly used to determine the quarantine period. The key contribution of this paper is the development of an R package that made the proposed method more accessible to epidemiologists. The authors estimated the serial intervals of outbreaks of 2009 influenza A/H1N1 and multiple COVID-19 variants in a range of populations. They should that their estimates are generally inline with parametrized estimations in the literature.

I have a few major concerns:

1. The authors needs to conduct a more thorough the literature review. For example, there are a few systematic reviews of serial interval estimations, especially for the COVID-19 pandemic; see, e.g., Rai et al. (2021) and Alena et al. (2021) below. Interval censored data have also been used in serial interval estimation, albeit using parametric methods; see, e.g., Nishiura (2020) below.

* Rai, B., Shukla, A., & Dwivedi, L. K. (2021). Estimates of serial interval for COVID-19: A systematic review and meta-analysis. Clinical epidemiology and global health, 9, 157-161.

* Alene, M., Yismaw, L., Assemie, M. A., Ketema, D. B., Gietaneh, W., & Birhan, T. Y. (2021). Serial interval and incubation period of COVID-19: a systematic review and meta-analysis. BMC infectious diseases, 21, 1-9.

* Nishiura, H., Linton, N. M., & Akhmetzhanov, A. R. (2020). Serial interval of novel coronavirus (COVID-19) infections. International journal of infectious diseases, 93, 284-286.

2. Infection pairs data collected in an early stage of an outbreak may exclude long serial intervals because of the short data collection period, see, e.g., Nishiura et al (2020) above. This may need to be considered in the method.

3. Clinical serial intervals, as stated in the introduction, may be negative, especially for COVID-19 or influenza, because of pre-symptomatic transmissions. The main consequence is that it may not be obvious that the infector is the patient in the pari with the earlier symptom onset date. How do we deal with the uncertainty of who the infector is? This may be further complicated

4. Nonparametric estimation is a useful way to validate parametric estimations, and it avoid the need to assume a distribution a priori. However, many applications requires the distribution itself. For example, the inference of the basic reproduction number from the exponential growth rate requires the calculation of the Laplace transform of the probability density function. For these applications, parametric estimation seems to be easier to work with. Can you explain how your method can be used in such calculations?

4. This method uses bootstrapping to estimate confidence intervals, which is a commonly used nonparametric method. However, it requires that the errors are i.i.d. yet, the serial intervals may not be i.i.d., because the transmissibility of a patient is typically not a constant. This can be demonstrated from the uncertainties in the figures, where the bands has a width roughly independent of the quantile. This may be quite different from parametric estimations. For example, this may affect the recommendation for quarantine periods.

Minor comments:

1. Although the authors give a comparison between their nonparametric estimates with parametric estimates in the literature, it would be interesting to conduct a more thorough comparison of various estimation methods in a table.

2. Page 4, line 2, by serial interval times do you mean the symptom onset times?

Reviewer #2: "Nonparametric serial interval estimation" by Gressani and Hens describes a nonparametric approach that applies the inverse CDF method to the empirical CDF of observed serial interval windows. The paper is very clearly written and easy to follow. The results seem promising overall. I think that some aspects of the writing should be expanded upon to clarify the nature of this contribution in light of prior and future work, and particularly to clarify whether the primary focus of this manuscript is Methodological (as I think it is). Since the authors have published methods to address similar problems in their reference 36 (https://academic.oup.com/aje/advance-article/doi/10.1093/aje/kwae192/7710093) it is a little curious that more direct comparison between the two methods is not made in the present manuscript, especially given their surface-level similarity. It probably behooves the authors to make some more explicit mention on this point.

It also seems to me that the overall focus of this paper is Methodological, rather than conveying new insights from the previously published data. However, the current Applications section (and the total lack of any dedicated subsection for simulation studies) raises the expectation that the paper's findings are primarily epidemiological, when in fact they are not. (As far as I can tell, all of the nonparametric estimates are in agreement with earlier results from parametric methods; this is communicated as an indication that the method is working correctly.) The main text ought to at least communicate the scenarios where the method succeeds or breaks down. It would be nice if there was a figure summarizing the results of, e.g. the Bias columns from Tables S2-S7 in the main text (or whatever statistic you think is most crucial). I suspect that more than one figure describing the simulation study results in the main text of the paper would greatly assist readers.

Minor comments:

At the end of the first paragraph in the Introduction, vague reference is made to "best practices" as articulated in [10]. If these can be consolidated into a few points, it would be helpful to readers for these to be described explicitly; or at least, to describe the best practices that your manuscript most closely aligns with

Also, in the second-to-last sentence of that first paragraph, "transmission interval" is a somewhat confusing or ambiguous term that, if I understand correctly, might just be replaced with the word "quantity"

Third paragraph of Introduction: The explanation for censoring here is nice, although I was not sure what "coarseness" referred to until reaching the end of the paragraph when you mention that symptoms are reported on a timescale of calendar days. For readers trying to work through this introduction on their first pass, it might be helpful if you mentioned the calendar time issue towards the beginning, and then expounded on coarseness in general.

Overall comment on the Introduction: I found myself wondering about the role of uncertainty in who-infects-whom. It seems that this method assumes that identities of the individuals involved in the transmission pairs are assumed known without error. I'm mindful that this might be an assumption made in earlier methods using parametric models as well, and don't at all intend to apply an inconsistent standard here, but think that the manuscript would benefit from some discussion on this point. Or, maybe the Conclusion section would be a better place to speculate on how the method might be extended in the future to accommodate errors in the who-infects-whom data.

Inverse-CDF pseudocode: I confess that line 4 of the pseudocode (Else return s* ... ) was not immediately obvious. Could you include a derivation of this, maybe just in the Supplement?

The bootstrap: is the size of the bootstrap samples, n, the same as the number of observed transmission pairs in D? If so, is there some particular reason for doing so?

Applications: While these results are all very promising, I could not help but think that organizing these into a table would be a much more efficient way to convey the information, and facilitate rapid comparison between your nonparametric method and earlier parametric methods. The results would also benefit from including some information about the different studies you obtained the data from. For instance, why are the serial interval estimates for H1N1 from New York so different from H1N1 from San Antonio? Did the studies come from schoolchildren of different ages, etc? I understand that it isn't too much work for a reader to examine the references themselves, but your article would be much more self-contained if you elaborated on this a little bit.

Section 3.3: Illness onset data for 2019-nCoV in Wuhan, China: It seems surprising that the CI95 for the earlier parametric model is so much wider than yours. I would have naively expected estimates from parametric models to be more precise than their counterparts from nonparametric methods. Do you have an explanation for this?

General comment on figures: I printed these on a black-and-white printer, but the ECDFS are all easily legible.

Conclusion: I found this a bit short, and was somewhat disappointed by the rather abrupt end of the paper. Would you care to comment on future extensions to this method that you would hope to implement in the future? Are there obvious next steps that other researchers can take to build on this work? What about elaborating on underlying assumptions of the method (like certainty in the identities of individuals involved in transmission pairs, or population structure)? Is there a distinction to be made between analysis of data collected in outbreak settings vs. endemic settings? What if the serial interval distribution changes over time (due to, say, evolution in the pathogen, implementation of widespread non-pharmaceutical interventions, or changes in age demographics)? Not all of these need to be addressed, but I think elaboration on some aspect of future directions should be included.

Supplementary material: Looking through the tables, I was struck by the pattern of CP_90 for q0.95. It looks like Scenario 1 is better than Scenarios 5 or 9, when I would have expected double-interval censoring to produce a worse result than single-interval censoring or no censoring. This (q0.95) is obviously going to be one of the hardest things to estimate, but is there any intuition to convey about why this is happening?

Reviewer #3: Context

-------

The serial interval (SI) of is defined as the time between symptom(s) onset of the infector and the symptoms onset of the infectee. This time can be negative since there nothing to stop the illness onset time of the infector to be greater than the onset time of the infectee. It is consider as a key quantity in infectious disease modelling since it is often used as a proxy of the generation interval.

When time intervals of illness onset between infectors and infectees are observed, then the most commonly method used to estimate the serial interval distribution is by fitting a parametric distribution (e.g. logNormal, Weibull, Gamma etc) to these data, assuming that they are a random sample from the population. Many of these methods have already been implemented in R packages.

A minimal requirement for SI estimation is to have data on symptom(s) onset times for the infector and infectee. However, serial interval data are often censored; either single, or doubly interval-censored.

This paper

-----------

This paper is concerned with estimating the distribution of the serial interval non-parametrically.

The basic idea behind this paper is to (i) view the data from a pair of infector-infectee as two observations from the serial interval distributions, (ii) obtain a continuous estimate of the CDF of the SI distribution by smoothing the empirical CDF using interpolation. One can draw samples from the serial interval distribution using the inverse-CDF method and then use bootstrap to estimate summary statistics of the SI distribution.

After a nice introduction to serial interval estimation, a simple estimator is proposed in Section 2, where in Section 3 application of the proposed method/estimator is done in several publicly availale dataset and compared to parametric alternativess.

Comments

--------

I have very much enjoyed reading this paper; it is very well written, well-structured and easy to read. The proposed estimator is very simple to implement and is already available in an R package (EpiLPS)

My main and only concern about the paper is its level of originality/innovation. It is not entirely clear from Section 2.2 but it appears to be either an application, or a straightforard extension, of the algorithm presented in [38] and [39]. In either case, I feel that this is below the bar for Plos Comp Biology.

Furthermore, in addition to the application of the algorithm to real infectious disease outbreak data, it would have been nice to have a simulation-study section to demonstrate the proposed algorithm under different scenarios (e.g. different sample sizes) and comparison against the true serial interval distributions.

The font sizes in some figures were pretty small.

**Have the authors made all data and (if applicable) computational code underlying the findings in their manuscript fully available?**

Reviewer #1: Yes

Reviewer #2: Yes

Reviewer #3: Yes

PLOS authors have the option to publish the peer review history of their article (what does this mean?). If published, this will include your full peer review and any attached files.

Reviewer #1: No

Reviewer #2: No

Reviewer #3: No

**Figure resubmission:**
---

## [Decision Letter · Decision Letter 1]

16 Jun 2025

PCOMPBIOL-D-24-01995R1

Nonparametric serial interval estimation with uniform mixtures

PLOS Computational Biology

Dear Dr. Gressani,

Thank you for submitting your manuscript to PLOS Computational Biology. After careful consideration, we feel that it has merit but does not fully meet PLOS Computational Biology's publication criteria as it currently stands. Therefore, we invite you to submit a revised version of the manuscript that addresses the points raised during the review process.

Please submit your revised manuscript within 30 days Aug 16 2025 11:59PM. If you will need more time than this to complete your revisions, please reply to this message or contact the journal office at ploscompbiol@plos.org. Please include the following items when submitting your revised manuscript:

We look forward to receiving your revised manuscript.

Kind regards,

Benjamin Peirce Holder, Ph.D.

Guest Editor

PLOS Computational Biology

Denise Kühnert

Section Editor

PLOS Computational Biology

**Additional Editor Comments :**

Dear Authors,

Thank you for your careful attention to the comments and suggestions by the reviewers and myself on the original manuscript, and for your preparation of an improved revised manuscript. Two of the original reviewers and I have read and commented on your revised manuscript (Reviewer #1 was not available for this re-review). Our suggestions and comments are relatively minor and should be able to be addressed in a "Minor Revision".

As before, I support the comments and suggestions provided by the reviewers, and would appreciate your best efforts to address them and the additional comments I have listed below.

Ben Holder

Guest Editor

* The new interpretation of your cdf through the uniform mixture model is a very nice addition. I would only suggest that the references [45,46] be given slightly more credit in your wording. In the original manuscript, your method was presented as an application of this prior work, but with the new manuscript these are listed simply as related works. I would recommend that your method is presented as an application and extension of that prior work, if that was indeed the origin.

* I would eliminate the phrase "interval width" to refer to the window of serial interval widths (line 123), because it could cause confuseion with the width of the serial interval itself. I think "window of serial interval", "window of serial interval width", or just "serial interval window" (used, e.g., in line 145) work well.

* What is the meaning of the "zero" in the inequalities in lines 120-121? Can it be eliminated, or should it appear in Figure 1?

* Figure 1 could provide a nice graphical visualization of the text in the first paragraph of 2.1, but to me it does not really present enough specific aid to the reader to be useful/included. Figure 1A is just a picture of calendars, but is not related to the timings. The number lines have no indication of the discreteness of days that leads to the coarseness. There is no attempt to present the rule for censoring that is given in lines 110-113. There is no attempt to register Figure 1C with Figure 1B, showing how the censored window is related to the reported values. If specifics like these could be added, without make the presentation too complicated, the figure would be much more useful.

* Also, in Figure 1D, it seems that two things are being identified: the definition of the serial interval, and its coarseness. Perhaps place the "Serial interval coarseness..." below the number line and title the panel something like "The serial interval and its coarseness"

* The set \mathcal{D} (lines 144-145, 182, 196-201) is unclear. It seems to me that you are using two different sets with the same name. One is a set of ordered pairs

\mathcal{D} = \left\{ ( s_{Li}, s_{Ri} ) \right\}_{i=1}^n

from which you sample the boostrap set (line 198). And the other is the union of all window endpoints (as defined in line 145), i.e., a simple set of the breakpoints from which you can take "a subset of \mathcal{D}" (line 182). I think this could be made a bit more clear, perhaps just sticking to the first set (the set of pairs), since the union of points really only seems to be referenced at line 182.

* You seem to be avoiding the explicit introduction and notation for the (uniform) pdf from which the cdf is obtained. It might be useful however to use this, e.g., in line 143:

f_{\mathcal{S}_i}(s) \sim \mathcal{U}(s_{iL}, s_{iR})

and then use this pdf within ordinary integrals over s for the definitions/derivations given in the equations between lines 178 and 179. This notation is more common than the RS integral and could make the presentation more accessible to a broader range of readers. It seems that the RS integral is only presented because notation for a pdf has not been introduced. But I leave it up to you to make this decision.

* In lines 223-227 there seems to be some inconsistency in how you are using \mathcal vs roman font to represent the distribution itself vs. the sample from that distribution.

* In Figure 3, there is no "A" or "B" (top/bottom?). Also there is no scale on the vertical axis... should there be, or does it not matter?

* I agree with the comments of Reviewer 2 about the Discussion. I think it could be also improved by beginning with a short overview paragraph of your results.

* A careful copy editing of the text (particularly new sections/paragraphs) is needed. A few typos appear at lines: 294, 301, 436.

**Reviewers' comments:**

Reviewer's Responses to Questions

Reviewer #2: The authors have made nice improvements to their manuscript and addressed the concerns I raised. I am glad to recommend acceptance with minor revisions to the writing.

I found the Discussion section a bit terse, overall. I think there might be room to elaborate more on the results about coarseness obscuring the SI distribution, and I think the limitation and future directions paragraphs could be expanded upon.

Very minor comments:

Laying out the best practices from [5] toward the end of the Introduction but making reference to those best practices in paragraph 1 reads a little bit disjointed (but thank you for adding the details!)

Check spelling in 3.1.3: The impact of coarseness - there are several typographic errors

The tables are a good improvement, but the captions can be totally self-contained to help readers navigate them more rapidly, i.e. should probably define what ESE, RMSE, etc. are within the captions, or at least in Table 1's caption

Do you need to say something about was assumed about coarsening in 3.2.1 and 3.2.2? It isn't immediately clear to me why you need to "apply" coarsening in 3.2.3 but not the others

Table 5: I am slightly confused by the apparently inconsistent shading of the rows

Reviewer #3: I would like to thank the authors for taking the time to respond to my questions -- having included a new section with a simulation study and presenting the non parametric estimation as a uniform mixture model has raised the novelty bar and it will be a useful addition to the litereture.

**Have the authors made all data and (if applicable) computational code underlying the findings in their manuscript fully available?**

Reviewer #2: Yes

Reviewer #3: Yes

PLOS authors have the option to publish the peer review history of their article (what does this mean?). If published, this will include your full peer review and any attached files.

Reviewer #2: No

Reviewer #3: No

**Figure resubmission:**
---

## [Decision Letter · Decision Letter 2]

18 Jul 2025

Dear Dr Gressani,

We are pleased to inform you that your manuscript 'Nonparametric serial interval estimation with uniform mixtures' has been provisionally accepted for publication in PLOS Computational Biology.

Best regards,

Benjamin Peirce Holder, Ph.D.

Guest Editor

PLOS Computational Biology

Denise Kühnert

Section Editor

PLOS Computational Biology

The reviewers and this guest editor appreciate the work the authors have done to address all concerns.

Reviewer's Responses to Questions

**Comments to the Authors:**

Reviewer #2: The authors have addressed my concerns and I congratulate them on a nice manuscript.

Reviewer #3: N/A

**Have the authors made all data and (if applicable) computational code underlying the findings in their manuscript fully available?**

Reviewer #2: Yes

Reviewer #3: None

PLOS authors have the option to publish the peer review history of their article (what does this mean?). If published, this will include your full peer review and any attached files.

Reviewer #2: No

Reviewer #3: No

---

## [Editor Report · Acceptance letter]

PCOMPBIOL-D-24-01995R2

Nonparametric serial interval estimation with uniform mixtures

Dear Dr Gressani,

I am pleased to inform you that your manuscript has been formally accepted for publication in PLOS Computational Biology. Your manuscript is now with our production department and you will be notified of the publication date in due course.

With kind regards,

Anita Estes
